# The geometric nature of weights in real complex networks

Antoine Allard[1,2], M. Ángeles Serrano[1,2,3], Guillermo García-Pérez[1,2] & Marián Boguñá[1,2]

The topology of many real complex networks has been conjectured to be embedded in hidden metric spaces, where distances between nodes encode their likelihood of being connected. Besides of providing a natural geometrical interpretation of their complex topologies, this hypothesis yields the recipe for sustainable Internet's routing protocols, sheds light on the hierarchical organization of biochemical pathways in cells, and allows for a rich characterization of the evolution of international trade. Here we present empirical evidence that this geometric interpretation also applies to the weighted organization of real complex networks. We introduce a very general and versatile model and use it to quantify the level of coupling between their topology, their weights and an underlying metric space. Our model accurately reproduces both their topology and their weights, and our results suggest that the formation of connections and the assignment of their magnitude are ruled by different processes.

[1] Departament de Física de la Matèria Condensada, Universitat de Barcelona, Martí i Franquès 1, E-08028 Barcelona, Spain. [2] Universitat de Barcelona Institute of Complex Systems (UBICS), Barcelona, Spain. [3] Institució Catalana de Recerca i Estudis Avançats (ICREA), Passeig Lluís Companys 23, E-08010 Barcelona, Spain. Correspondence and requests for materials should be addressed to M.B. (email: marian.boguna@ub.edu).

Most of the complexity of networks is encoded into the intricate topology of the interactions among their components and into the layout of the intensities associated to such interactions (the weights). Interestingly, weights are coupled in a non-trivial way to the binary network topology, playing a central role in their structural organization, function and dynamics[1]. For instance, the quantification of the rich-club effect in real weighted networks, in sharp contrast to results in unweighted representations, unveils the formation of alliances in multipolarized environments or the lack of cohesion even in the presence of rich-club ordering[2]. Similarly, the propagation of emergent diseases in the international airports network is intimately linked to the number of passengers flying from one airport to the other[3]. A shift towards a paradigm of weighted networks is therefore in order to fully understand the behaviour and evolution of complex networks. However, advances in this area have been limited by the extreme heterogeneity and fluctuations that typically characterize the distribution of weights.

Meanwhile, complex networks[4,5] have been conjectured to be embedded in hidden metric spaces, in which distances among nodes encode a balance between their similarity and popularity and, thus, determine their likelihood of being connected[6]. This hypothesis, combined with a suitable underlying space, has offered a geometric interpretation of the complex topologies observed in real networks, including scale-free degree distributions, the small-world effect, strong clustering, community structure and self-similarity. A metric space under-lying complex networks can also explain their efficient inter-node communication without knowledge of the complete structure[7,8]. Moreover, it has been shown that for networks whose degree distribution is scale free, the natural geometry of their underlying metric space is hyperbolic[9–14]. All these results have then been used to propose geometric models for real growing networks that reproduce their evolution and in which preferential attachment emerges from local optimization principles[15,16]. Finally, mapping real complex networks into a hidden metric space has yielded a sustainable solution to the scaling limitations of the Internet[17], has shed light on the hierarchical organization of biochemical pathways in cells[18], and has allowed a rich characterization of the evolution of international trade over 14 decades[19].

In real weighted networks, weights are coupled to the binary topology in a non-trivial way. This is manifested, for instance, in a non-linear relation between the strength of a node $s$ (the sum of the total weight attached to it) and its degree $k$ of the form $s \sim k^\eta$ (refs 1,20,21). However, the relation between the layout of weights and the geometry underlying the network is unclear. The reason being that, even if the existence of a link depends on the metric distance between the nodes, there is no reason, a priori, to expect that the same distance will affect its weight. For instance, in the airports network, the decision to set-up a link between two cities depends on the airline companies operating at the two airports, a process affected by geopolitic and economic costs, and by the expected flow of passengers that would eventually compensate such costs. However, once the connection is established, its weight is determined by the aggregation of the individual decisions of people using it, a process that may be affected by a different cost function.

In this paper, we present empirical evidence on the metric nature of weights in real biological, economic and transportation networks (see Methods for a description of the data sets), which suggests that the hidden/latent geometry paradigm can be extended to weighted complex networks. We then propose a general class of weighted networks embedded in hidden metric spaces that accurately reproduces many properties observed in real weighted networks. This model has the critical ability to fix the degree–strength distribution independently of the coupling of the topology and weighted organization with the metric space. It is therefore possible to isolate, and thus directly study, the effect of the coupling between the metric space and the weighted organization of real weighted networks. In fact, our results unveil that in some systems these couplings are uncorrelated, which in turn suggests that the formation of connections and the assignment of their magnitude might be ruled by different processes. Our empirical findings, combined with our new class of models, open the path towards the use of information encoded in the weights of the links to find more accurate embeddings of real networks, which in turn will improve the detection of communities, the prediction of missing links and provide estimates for the weights of such missing links.

## Results

**Interplay between weights and triangles in real networks.** Clustering, as a reflection of the triangle inequality, is the key topological property coupling the bare topology of a complex system and its effective underlying metric space[6]. In this context, the triangle inequality stipulates that if nodes A and B are close, and nodes A and C are also close, we expect nodes B and C to be close as well; triangles are therefore more likely to exist between nodes that are nearby. Consequently, we expect that if the weights of connections depend on the distance between the connected nodes in the underlying metric space, they should be quantitatively different depending on the clustering properties of the connections. However, weights and clustering are known to be strongly influenced by the degrees of end point nodes[1,20,22], which prevents from a direct detection of the metric properties of weights due to the typical heterogeneity in the degrees of nodes in real networks. Thus, to compare links on an equal footing, we define the normalized weight of an existing link connecting nodes $i$ and $j$ as $\omega_{ij}^{\mathrm{norm}} = \omega_{ij}/\bar{\omega}(k_i k_j)$, where $\bar{\omega}(kk')$ is the average weight of links as a function of the product of degrees of their end point nodes. By doing so, we decouple the weights and the topology, leaving the normalized weights seemingly randomly fluctuating around 1 (see uniform sampling on Fig. 1).

Figure 1 shows, however, that these fluctuations are not uniform as links involved in triangles tend to have larger normalized weights than the average link. Indeed, in some cases the difference can reach >30%. Sampling links over triangles is equivalent to sampling links proportionally to their multiplicity $m$ (the number of triangles to which a link participates). Therefore, the results in Fig. 1 indicate that $\omega^{\mathrm{norm}}$ and $m$ are positively correlated variables, as corroborated by their Pearson correlation coefficient (Supplementary Table 1). In ref. 22, the authors also found local correlations between the multiplicity of links and the weights for different real networks. However, note that in that study weights were not normalized to discount the effects of the heterogeneity in the degrees of the nodes, so that the detected weighted organization cannot be taken as a signature of underlying metric properties.

Since triangles are a reflection of the triangle inequality in the underlying metric space, we expect nodes forming triangles to be close to one another. Thus, the higher average normalized weight observed on triangles strongly suggests a metric nature of weights, which is not a trivial consequence of the relation between weights and topology. This leads us to formulate the hypothesis that the same underlying metric space ruling the

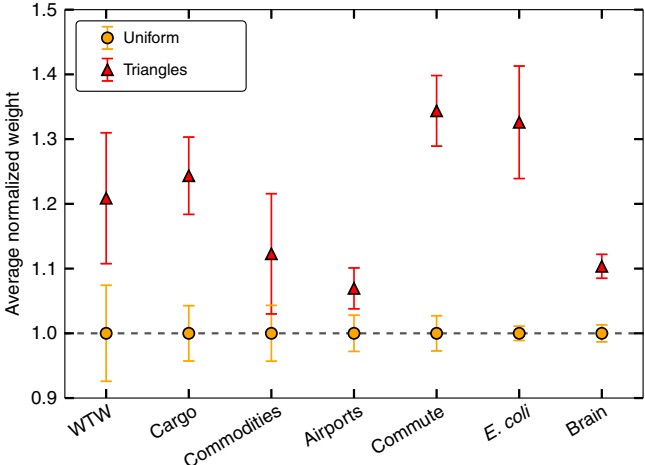

**Figure 1 | Geometric nature of weights.** Comparison of the average normalized weights in the network (yellow circles) with the one measured by sampling links over triangles (red triangles) for the empirical data sets analysed. The error bars correspond to an estimate of the s.d. of the average value due to the finite size of the samples and are computed as $\sqrt{\text{Var}[\omega^{\text{norm}}]/L}$, where $\text{Var}[\omega^{\text{norm}}]$ is the variance of the normalized weights sampled uniformly or via the triangles, and $L$ is the number of links.

network topology—inducing the existence of strong clustering as a reflection of the triangle inequality in the underlying geometry—is also inducing the observed correlation between $\omega^{\text{norm}}$ and $m$. To prove this, we develop a realistic model of geometric weighted random networks, which allows us to estimate the coupling between weights and geometry in real networks.

**A geometric model of weighted networks.** Many models have been proposed to generate weighted networks. Among them, growing network models[23–30] and the maximum-entropy class of models[31–35]. However, none of them is general enough to reproduce simultaneously the topology and weighted structure of real weighted complex networks. We introduce a new model based on a class of random networks with hidden variables embedded in a metric space[6,7] that overcomes these limitations. In this model, $N$ nodes are uniformly distributed with constant density $\delta$ in a $D$-dimensional homogeneous and isotropic metric space (Supplementary Methods), and are assigned a hidden variable $\kappa$ according to the probability density function (pdf) $\rho(\kappa)$. Two nodes with hidden variables $\kappa$ and $\kappa'$ separated by a metric distance $d$ are connected with a probability

$$\text{Prob}(\kappa, \kappa', d) = p(\chi), \quad \text{and} \quad \chi = \frac{d}{(\mu\kappa\kappa')^{1/D}}, \qquad (1)$$

where $\mu > 0$ is a free parameter fixing the average degree and $p(\chi)$ is an arbitrary positive function taking values within the interval (0, 1). The free parameter $\mu$ can be chosen such that $\bar{k}(\kappa) = \kappa$. Hence, $\kappa$ corresponds to the expected degree of nodes, so the degree distribution can be specified through the pdf $\rho(\kappa)$, regardless of the specific form of $p(\chi)$ (Supplementary Methods). The freedom in the choice of $p(\chi)$ allows us to tune the level of coupling between the topology of the networks and the metric space, which in turn allows us to control many properties such as the clustering coefficient and the navigability[6,8].

To generate weighted networks, a second hidden variable $\sigma$ is associated to each node. This new hidden variable can be

correlated with $\kappa$ so, hereafter, we assume that the pair of hidden variables ($\kappa$, $\sigma$) associated with the same node are drawn from the joint pdf $\rho(\kappa, \sigma)$. The weight of an existing link between two nodes with hidden variables $\kappa_i$, $\sigma_i$, $\kappa_j$ and $\sigma_j$, respectively, and at a metric distance $d_{ij}$ is given by

$$\omega_{ij} = \epsilon_{ij} \frac{\nu\sigma_i\sigma_j}{(\kappa_i\kappa_j)^{1 - \alpha/D} d_{ij}^\alpha} \qquad (2)$$

with $\nu > 0$ and $0 \leq \alpha < D$ and where $\epsilon$ is a positive random variable drawn from the pdf $f(\epsilon)$. Notice that $\alpha$ dictates a trade-off between the contribution of degrees and geometry to weights. If $\alpha = 0$ weights are independent of the underlying metric space and maximally dependent on degrees, while $\alpha = D$ implies that weights are maximally coupled to the underlying metric space with no direct contribution of the degrees. Equation (2) constitutes the keystone of our model. Indeed, as shown in the Supplementary Methods, the form of equation (2) is the only one ensuring that $\bar{s}(\sigma) \propto \sigma$. The free parameter $\nu$ can then always be chosen such that $\bar{s}(\sigma) = \sigma$. The new hidden variable $\sigma$ can therefore be interpreted as the expected strength of a node, and the joint pdf $\rho(\kappa, \sigma) = \rho(\kappa)\rho(\sigma|\kappa)$ controls the correlation between degrees and strengths in the network. Indeed, as shown in the Supplementary Methods, the average strength of nodes with a given degree, $\bar{s}(k)$, relates to the first moment of the conditional pdf $\rho(\sigma|\kappa)$, $\bar{\sigma}(\kappa)$, so that when $\lim_{\kappa\to\infty}\bar{\sigma}(\kappa) = \infty$ then $\bar{s}(k) \sim \bar{\sigma}(\kappa)$.

The relations $\bar{k}(\kappa) = \kappa$ and $\bar{s}(\sigma) = \sigma$—and consequently the relation between $\rho(\kappa, \sigma)$ and the degree–strength distribution—hold independently of the specific form of the connection probability $p(\chi)$ and of the noise distribution $f(\epsilon)$. Besides conferring great versatility to our model, this conveys a degree of control over the weight distribution that is independent of the specification of degrees and strengths and, more importantly, opens the possibility to measure the metric properties of complex weighted networks.

To use the model in the context of real weighted networks, we choose the circle $\mathbb{S}^1$ of radius $R = N/2\pi$ to be the underlying geometry, that is, $D = 1$, over which $N$ nodes are uniformly distributed[6]. Distances among nodes are measured in terms of arc lengths, that is, two nodes with angular positions $\theta$ and $\theta'$ are at a distance $d(\theta, \theta') = R\Delta\theta$, where $\Delta\theta = \pi - |\pi - |\theta - \theta'||$. The connection probability is set to

$$p(\chi) = \frac{1}{1 + \chi^\beta} \quad \text{with} \quad \chi = \frac{d}{\mu\kappa\kappa'}, \qquad (3)$$

where $\beta > 1$ is a free parameter that can be used to tune the clustering and quantifies the level of coupling between the network topology and the metric space. Equation (3) casts the ensemble of networks generated by the model into exponential random networks[9]: networks that are maximally random given the constraints imposed by the free parameters (that is, $\rho(\kappa)$ and $\beta$). To obtain a scale-free degree distribution, hidden variables $\kappa$ are distributed according to $\rho(\kappa) \propto \kappa^{-\gamma}$ with $\kappa_0 < \kappa < \kappa_c$ and $\gamma > 1$.

Weights are assigned on top of the topology generated by the model. The noise distribution $f(\epsilon)$ is chosen to be a gamma distribution of average $\langle\epsilon\rangle = 1$ with a given second moment $\langle\epsilon^2\rangle$. Finally, to control the correlation between strength and degree and, therefore, to tune the strength distribution, we assume a deterministic relation between hidden variables $\sigma$ and $\kappa$ of the form $\sigma = a\kappa^\eta$, as observed in real complex networks[1,20,21], yielding $\bar{s}(k) \sim a k^\eta$ (Supplementary Methods). Notice that the relation between average strength and degree in the previous expression is totally independent of the underlying metric space, which implies that the strength distribution

scales as $P(s) \sim s^{-\xi}$ for $s \gg 1$ with $\xi = (\gamma + \eta - 1)/\eta$. All these theoretical predictions and the ones derived in the Supplementary Methods are confirmed in Supplementary Fig. 1.

**Hidden metric spaces underlying real weighted networks.**
At the beginning of this section, we showed that the normalized weights of links participating in triangles are higher, thus suggesting a coupling between the weighted organization of real weighted complex networks and an underlying metric space. We then presented a model that has the critical ability to fix the joint degree–strength distribution, while independently varying the level of coupling between the weights and the metric space (parameter $\alpha$). This opens the way to a definite proof of the geometric nature of weights in real complex networks, which inevitably must involve the triangle inequality: the most fundamental property of any metric space.

For unweighted networks, a direct verification of the triangle inequality based on the topology without an embedding in a metric space is not possible, due to the probabilistic nature of the relationship between the binary structure and the distance between nodes. In contrast, weights do contain information about their distances in the metric space (via equation (2)) such that a direct verification of the triangle inequality is possible. To ensure that the metric properties of triples in the network are in correspondence to the metric properties of the corresponding triangles in the underlying space, only triples of nodes forming triangles in the network are taken into account to evaluate the triangle inequality. There are however two main challenges when one tries to apply this methodology. The first one is related to the fact that connections in the weighted $\mathbb{S}^1$ model depend not only on angular distances but also on hidden degrees, such that we need a purely geometrical formulation of the weighted hidden metric space network model, in which angular distances and degrees are combined into a single distance measure. The second issue is related to the intrinsic noise present in the system due to the stochastic nature of the processes conforming it, which may blur the evaluation of the triangle inequality. Below, we propose a way to overcome these two issues.

First, as shown in ref. 9, the model described by equation (1) is equivalent, in the one-dimensional case, to a purely geometric model where nodes are embedded within a disk of radius $R$ in the hyperbolic plane of constant curvature $-1$. Indeed, by mapping the hidden variable $\kappa$ to a radial coordinate $r$ as follows

$$r = R - 2\ln\left[\frac{\kappa}{\kappa_0}\right] \quad \text{with} \quad R = 2\ln\left[\frac{N}{\mu\pi\kappa_0^2}\right] \quad (4)$$

and keeping the same angular coordinates, the connection probability equation (1) can be written as

$$p\left(\frac{d}{\mu\kappa\kappa'}\right) = p\left(e^{\frac{1}{2}(x-R)}\right) \quad (5)$$

where $x = r + r' + 2\ln\frac{\Delta\theta}{2}$ is a very good approximation of the hyperbolic distance between two points with radial coordinates $r$ and $r'$, and angular separation $\Delta\theta$. In this framework, networks generated with our model are geometric random networks in the hyperbolic plane, a geometry in which the triangle inequality must hold. To test the triangle inequality, we therefore select nodes participating in topological triangles in the network and measure the hyperbolic distance between them.

The purely geometric interpretation of our model given by equation (5) further illustrates the reasons for which a metric space implies a non-vanishing clustering even in the thermodynamical limit. As stated at the beginning of this section, the triangle inequality—a fundamental property of any metric space, including the hyperbolic plane—stipulates that whenever point A is close to point B and point B is close to point C, then points A and C are also close. Consequently, the notion of 'closeness' extends well beyond pairwise comparisons and is integrated 'at once' in the positions in the metric space. This implies that many-body interactions emerge from pairwise interactions, such as the connection probability given by equation (3). Given that nearby nodes are likely to be connected, clustering is a direct consequence of such many-body interactions; any triad of close nodes are likely to form a triangle, independently of the size of the disk, and therefore of the total number of nodes.

By using the mapping given by equation (4), with $D = 1$ equation (2) becomes

$$\omega_{ij} = \epsilon_{ij} \frac{\nu}{\mu^\alpha} \frac{\sigma_i \sigma_j}{\kappa_i \kappa_j} e^{-\frac{\alpha}{2}(x_{ij} - R)}, \quad (6)$$

from which we can isolate the hyperbolic distance, $x_{ij}$, between nodes $i$ and $j$. The triangle inequality, $x_{ij} + x_{jk} \geq x_{ik}$, then becomes

$$\ln\left[\frac{\omega_{ij}\omega_{jk}}{\omega_{ik}}\left(\frac{\kappa_j}{\sigma_j}\right)^2\right] + \ln\left[\frac{\mu^\alpha}{\nu}\right] - \frac{1}{2}\alpha R \leq \ln\left[\frac{\epsilon_{ij}\epsilon_{jk}}{\epsilon_{ik}}\right]. \quad (7)$$

The first term in the left hand side of this inequality is a function of the actual weights and network topology and, thus, can be empirically estimated in any network. The next two terms on the left hand side have an explicit dependence on the parameter $\alpha$. The term in the right hand side is a noise term whose mean value is close to zero.

Let us first assume that this noise term is zero. In synthetic weighted networks, the inequality should hold approximately for

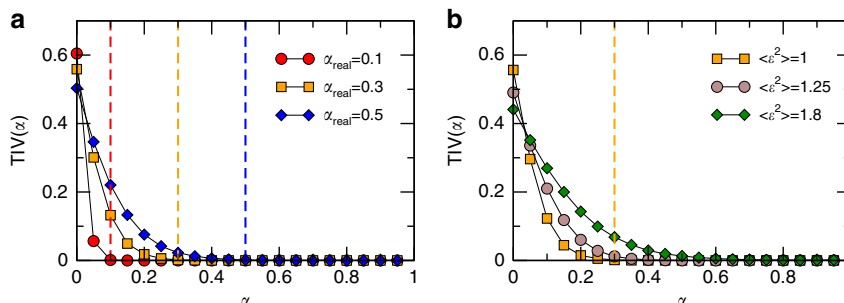

**Figure 2 | Triangle inequality violation in synthetic networks.** Fraction of violations of the triangle inequality, TIV($\alpha$) (that is, equation (7)) (**a**) without noise and different values of $\alpha_{real}$, and (**b**) with a fixed value of $\alpha_{real} = 0.3$ and different values of the noise $\langle \epsilon^2 \rangle$. In both cases, the topology is the same, with $\gamma = 2.5$, $\beta = 2$, $\eta = 1$ and $N = 10^4$. The vertical dashed lines indicate the values of $\alpha_{real}$ used to generate the different networks.

any value of $\alpha$ in equation (7) equal to or larger than the value of $\alpha_{real}$ used to assign weights in the network. Note that it may not hold exactly even when $\alpha$ is greater than its real value due to the inherent noise in the estimation of the hidden variables $\kappa$ and $\sigma$ in equation (7), as well as the global parameters $\mu$ and $v$ (note that whenever we set $\bar{s}(\sigma) = \sigma$, parameter $v$ becomes a function of $\alpha$; see Supplementary Methods). To minimize such uncertainty, we choose $\sigma = a\kappa^\eta$ and approximate $\kappa$ by the degree of nodes. We propose to consider $\alpha$ in equation (7) as a free parameter and to measure the triangle inequality violation spectrum, TIV($\alpha$), defined as the fraction of violations of the triangle inequality (triangles for which the left hand side of equation (7) is positive). In the absence of noise, TIV($\alpha$) should take a very small value when $\alpha \geq \alpha_{real}$ if the weighted structure of the network is congruent with the existence of an underlying metric space. In Fig. 2a, we show TIV($\alpha$) for synthetic networks generated with the model with different values of $\alpha_{real}$. As expected, the curves fall rapidly precisely at $\alpha \gtrsim \alpha_{real}$, indicated by the dashed vertical lines.

In real situations, however, noise is typically present and has an impact on TIV($\alpha$). Indeed, Fig. 2b shows its behaviour for a fixed value of $\alpha_{real}$ and different values of the noise $\langle \epsilon^2 \rangle$. This implies that we need an independent measure of the noise to infer the value of $\alpha_{real}$ from the spectrum TIV($\alpha$). For this purpose, we use the square of the coefficient of variation of the strength, which depends linearly on the noise $\langle \epsilon^2 \rangle$ (Supplementary Methods). Combining these observations, we propose a

procedure to infer the value of $\alpha_{real}$ for any real complex network based on the empirical TIV($\alpha$). The method is described in details in the Supplementary Methods.

Figure 3a,b shows the TIV($\alpha$) curves for the real networks and the same curves for synthetic networks generated by our model using the inferred $\alpha_{real}$ to be maximally congruent with the real data. In all cases, we find a very good agreement between theory and observations, which suggests a coupling with a hidden metric space as a highly plausible explanation of the observed weighted organization. Note that the increase of TIV($\alpha$) for $\alpha \sim 1$ is an expected artefact of equation (7) (Supplementary Methods). Figure 3c shows the values of $\beta$ (coupling topology and metric space) and $\alpha_{real}$ (coupling weights and metric space) inferred by our method. Notice that, except for the US airports network, $\alpha_{real}$ is always $> 0.40$, which indicates a clear and strong coupling between weights and the hidden underlying geometry. We also generated synthetic networks with the inferred parameters and confronted their topological and weighted properties against those of their real counterparts (see Fig. 4 and Supplementary Methods for other networks and a comparison with other models). In all cases, the agreement between the model and the real networks is excellent. Remarkably, in the case of the weight distribution and disparity measure, such agreement is only achieved with the empirical value of $\alpha_{real}$ found via the test of the triangle inequality.

Finally, we considered the networks for which an embedding of the binary structure was available and rescaled each weight by a

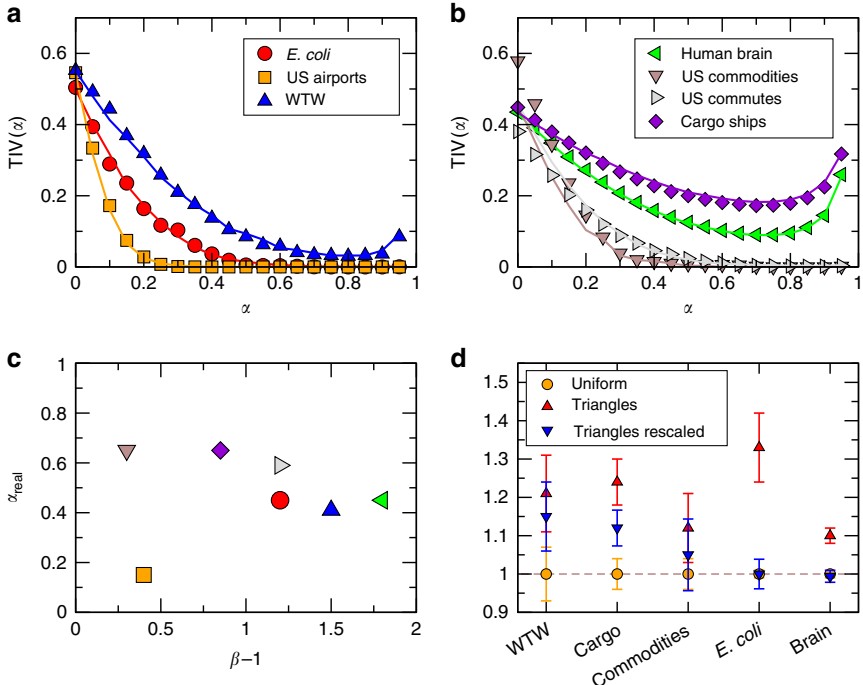

**Figure 3 | Triangle inequality violation in real networks.** Triangle inequality violation curves (**a,b**) for all real networks considered in this study (symbols). Solid lines correspond to their model counterparts with the model parameters in Supplementary Table 1. (**c**) Inferred values of the coupling parameter $\alpha_{real}$ versus $\beta - 1$. The numerical values of these parameters can be found in Supplementary Table 1. (**d**) Average normalized weights in the network (yellow circles) with the one measured by sampling links over triangles (red triangles) for most of the empirical data sets analysed. The error bars correspond to an estimate of the s.d. of the average value due to the finite size of the samples (see the caption of Fig. 1 for details). The blue inverted triangles correspond to the red triangles, but where the weights were rescaled by the factor $e^{-\alpha_{real}x_{ij}/2}$ to take into account the coupling between the weights and the hidden metric space. The airports and commuting networks could not be embedded into a metric space using current state-of-the-art methodology[17,46] due to atypical topological features and are therefore not reproduced here. These atypical features refer to a power-law degree distribution with an exponent below 2 in the case of the US airports network and a short-range repulsion effect in the connection probability for the commute network (that is, people rarely commute from one suburb to another but rather commute from one suburb to the major city in the area). This does not affect our general theory but rather prevent the state-of-the-art embedding algorithms to provide us with an embedding of these two networks.

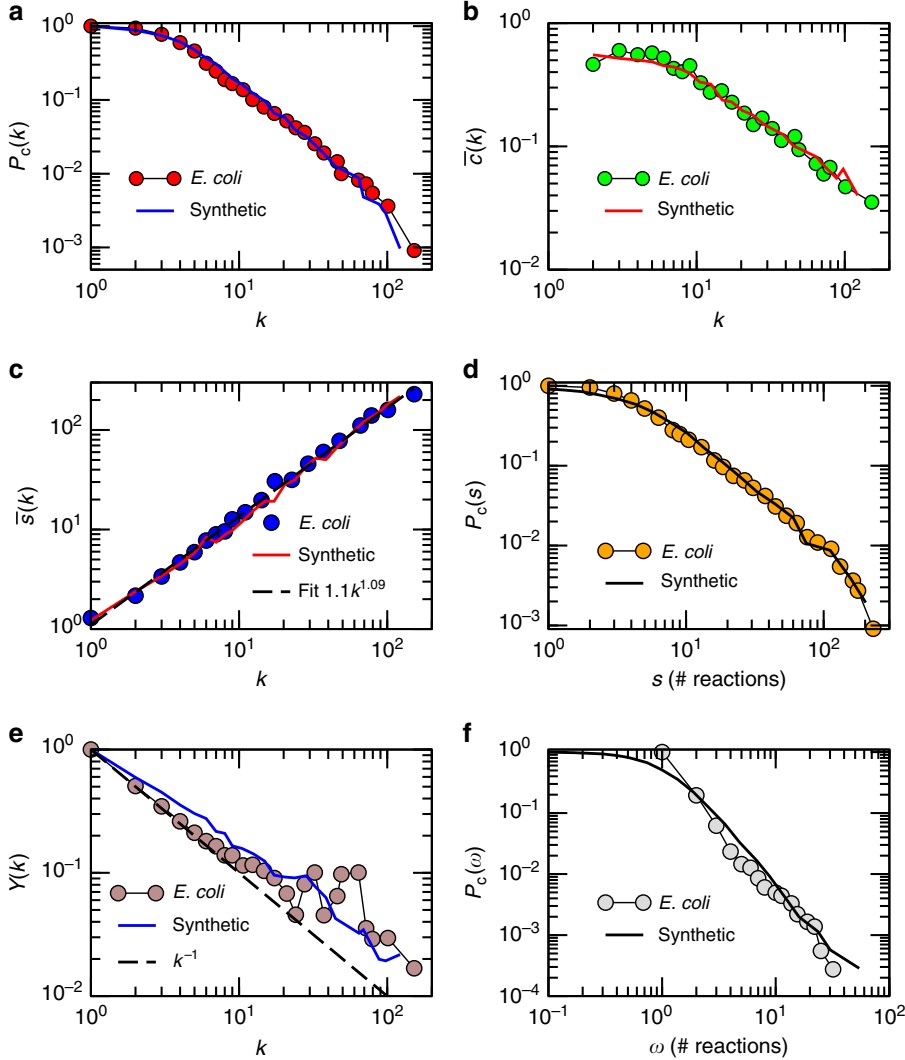

**Figure 4 | Model versus real network.** Comparison between topological and weighted properties of the iJO1366 *E. Coli* metabolic network (symbols) and a synthetic network generated by the model with the parameters given in Supplementary Table 1 (solid lines). (**a**) Complementary cumulative degree distribution. (**b**) Degree-dependent clustering coefficient. (**c**) Average strength of nodes of degree $k$. (**d**) Complementary cumulative strength distribution. (**e**) Disparity of nodes as a function of their degree (Methods). (**f**) Complementary cumulative weight distribution of links.

factor $e^{-\alpha_{real}x_{ij}/2}$ (equation (6)), where $x_{ij}$ is the hyperbolic distance between nodes in the embeddings[17]. We then normalized and sampled the weights as in Fig. 1 and the results are shown in Fig. 3d. Strikingly, we see that the gap observed in Fig. 1 completely disappears in some of the networks or is significantly reduced in others. While the remaining gaps may be due to imprecisions in the embedding (the embedding procedure cannot take into account the information contained in the weights yet), these results nevertheless add their voice to the evidence pointing towards the geometric nature of the weights in real complex networks.

## Discussion

The metric character of many real complex networks—in which clustering is a direct consequence of the triangle inequality—has long been established. However, the metric nature of their weighted organization still remained an open question. In this paper, we provided strong empirical evidence for the metric origin of the weighted architecture of real complex networks from very different domains. Our results suggest that the same

underlying metric space ruling the network topology also shapes its weighted organization. It is important to notice that the distances between nodes implied by this metric space does not necessarily correspond to geographic distances (for example, distances between ports on the Earth), but are rather abstract and effective distances encoding several factors affecting the existence of connections and their intensity.

To account for these empirical findings, we proposed a very general model capable of reproducing the coupling with the metric space in a very simple and elegant way. This model allows us to fix the local properties of the nodes—their joint degree–strength distribution—while varying the coupling of the topology and, independently, of the weights with the hidden metric space. This critical property permits us to gauge quantitatively the effect of the metric space in real systems. In the case of the US airports network, we found quite remarkably that while the coupling between the topology and the metric space is relatively strong, the coupling at the weighted level is quite weak. This strengthens the hypothesis that in some systems the formation of weights and topology obey different dynamics. Contrarily, we found strong coupling, both at the

**Table 1 | Overview of the considered real-world network data.**

| Name | Type | Nodes | Weights | $\gamma$ | $\eta$ | $\langle k \rangle$ | $N$ |
|---|---|---|---|---|---|---|---|
| World Trade | Economic | Countries | US dollars | 2.42 | 1.6 | 5.8 | 189 |
| Cargo ships | Transportation | Ports | Shipping journeys | 2.03 | 1.05 | 10.4 | 834 |
| US commodities | Economic | Economic sectors | US dollars | 2.46 | 1.22 | 5.8 | 376 |
| US airports | Transportation | Airports | Passengers | 1.76 | 1.7 | 8.6 | 884 |
| US commute | Transportation | Counties | People | 4.31 | 2.02 | 4.3 | 3109 |
| *E. coli* | Biological | Metabolites | Common reactions | 2.52 | 1.1 | 6.6 | 1100 |
| Human brain | Biological | Brain regions | Connection density | 7.14 | 0.86 | 24.1 | 501 |

Details for each data set can be found in Methods.

topological and weighted levels, even in networks that are not embedded in any obvious metric space like the metabolism of *Escherichia coli*, a system of metabolic reactions for which the hidden geometry is elucidated as a biochemical affinity space. This fact provides yet another empirical evidence towards the existence of hidden metric spaces shaping the architecture of these systems and, more generally, of real complex networks[6].

Our framework can be understood as a new generation of gravity models applicable to very different domains, including Biology, Information and Communication Technologies, and Social Systems. Indeed, equation (2) is a novel generalization of this concept to the case of weighted networks, where

$$\frac{\sigma}{\kappa^{1-\alpha/D}} \qquad (8)$$

plays the role of the 'mass' of nodes and ensures that, once the network has been assembled, nodes have expected degree and strength $\kappa$ and $\sigma$, respectively. Current gravity models predict the volume of flows between elements, but cannot explain the observed topology of the interactions among them, as shown in works for the world trade web[36]. Our contribution overcomes this limitation and offers a gravity model that can reproduce both the existence and the intensity of interactions. This opens a new line of theoretic research on the coupling between topology, weighted structure and geometry in complex networks.

Furthermore, our work opens the possibility to use information encoded in the weights of the links to find more accurate embeddings of real networks. Such improved embeddings are expected to allow the detection of communities or of missing links and to provide estimates of the weights of such missing links[37–39]. They can also be extremely helpful to implement navigation and searching protocols, such as greedy routing, which take into account not only the existence of connections but also their intensity.

In perspective, the hidden metric space weighted model and the maps of real complex systems that it will enable will lead to a deeper understanding of the interplay between the structure and function of real networks, and will offer insights on the impact they have on the dynamical processes they support and on their own evolutionary dynamics.

## Methods
**Empirical data sets.** In addition to the details given in Table 1, we provide further information and references about the real complex networks used in this paper.

The world trade web describes significant trade exchanges between countries in 2013. The corresponding weights are trade volumes between pairs in USD[19].

The international network of global cargo ship movements consists of the number of shipping journeys between pairs of major commercial ports in the world in 2007 (ref. 40).

The commodities network corresponds to the flows of the goods and services in millions of USD between industrial sectors in the United States in 2007 (ref. 41).

The airports network indicates the number of passengers that flew between pairs of airports in the United States in 2013. Data are freely available at the website of the US Bureau of Transportation Statistics (transtats.bts.gov).

The commuting network reflects the daily flow of commuters between counties in the United States in 2000 (ref. 41).

Weights in the metabolic network of the bacteria *E. Coli* K-12 MG1655 consist of the number of different metabolic reactions, in which two metabolites participate[18,42].

Weights in the human brain network correspond to the density of anatomical connections between subregions of the human brain as detected via diffusion tensor imaging[43].

Except for the metabolic and human brain networks, all networks were filtered using the disparity filter defined in ref. 44 to preserve the most statistically significant connections. Many real weighted networks are generated from data by using a very broad definition of what constitutes a significant connection. This results in networks with huge average degrees and in which many links are noisy and weakly related to the overall functionality of the network. For instance, the US airports network contains links due to private flights (of the order of 10 passengers per year), which obviously follow different patterns of connection than the regular commercial airlines. Another interesting example is the world trade web, in which many trade interactions amount for less than one million dollars and are extremely volatile, appearing and disappearing from year to year. Indeed, it has been shown in ref. 19 that removing these noisy connections yields a significantly more congruent topology with real economic factors, such as the gross domestic product.

**Disparity.** The disparity quantifies the local heterogeneity of the weights attached to a given node and is defined as

$$Y_i = \sum_j \left( \frac{\omega_{ij}}{s_i} \right)^2, \qquad (9)$$

where $\omega_{ij}$ is the weight of the link between nodes $i$ and $j$ ($\omega_{ij} = 0$ if there is no link) and $s_i = \sum_j \omega_{ij}$ (ref. 45). From this definition, we see that the disparity scales as $Y_i \sim k_i^{-1}$ whenever the weights are roughly homogeneously distributed among the links. Conversely, whenever the disparity decreases slower than $k_i^{-1}$ implies that weights are heterogeneous and that the large strength of a node is due to a handful of links with large weights.

**Data availability.** Codes and data supporting the findings of this study are available from the corresponding author on request.

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

## Acknowledgements

We acknowledge support from the James S. McDonnell Foundation Scholar Award in Complex Systems; the Fonds de recherche du Québec—Nature et technologies; the ICREA Academia prize, funded by the Generalitat de Catalunya; the MINECO project no.FIS2013-47282-C2-1-P; and the Generalitat de Catalunya grant no.2014SGR608.

## Author contributions

A.A., M.Á.S., G.G.-P. and M.B. contributed to the design and implementation of the research, to the analysis of the results and to the writing of the manuscript.

## Additional information

**Competing financial interests:** The authors declare no competing financial interests.

**Publisher's note**: 

