## [Peer Review File · Nature Communications]

Reviewers' comments:

Reviewer #1 (Remarks to the Author):

A) Summary of the key results:

This paper extends the notion of geometric embedding of networks to the case of weighted networks. It introduces a model with coupled strengths and degrees and explicit embedding in a geometric space. It compares the model results with empirical network data. Although the results are interesting and the model is a new one, the approach is too specific and too many arbitrary assumptions are made to arrive to the final results. This paper is not of immediate interest to a broad audience.

I therefore do not recommend publication in Nature Communication.

B) Originality and interest:

The problem of modelling both weights and topology of real networks is not new, but more than a decade old. It dates back to when the first evidence of coupling between topology and weights has been provided (see Barrat, Alain, et al. "The architecture of complex weighted networks." Proceedings of the National Academy of Sciences of the United States of America 101.11 (2004): 3747-3752, and Barrat, Alain, Marc Barthélemy, and Alessandro Vespignani. "Weighted evolving networks: coupling topology and weight dynamics." Physical review letters 92.22 (2004): 228701.)

The main original contribution of this paper is a very specific model of geometrically embedded networks, where the weight structure is induced by the embedding.

However, the claimed generality of the approach is questionable and the model is too specific to appeal to a broad audience (see point C below).

Even for a specialized audience of network scientists, this paper is too restrictive to attract general interest. It provides an ad hoc mechanism that is unlikely to stimulate significant new research at a general level.

C) Data & methodology:

The approach is valid in itself, but the claim of being the most flexible and appropriate one is unjustified. The key assumption of the model, eq.(2), is a highly arbitrary choice. It relies on the assumption that there are two hidden variables per node, controlling for the expected degree and strength of that node respectively. Although the authors show that eq.(2) is the only one ensuring that the expected strength coincides with the associated hidden variable, this is an arbitrary and questionable criterion. Hidden variables are introduced precisely in order to 'explain' the expected properties (like strengths and degrees) via a hidden or underlying quantity which influences such properties, usually in a highly nonlinear way. Requiring the hidden variable to coincide with the expected value of the property it is controlling for is an unnecessary restriction.

The claim of generality is therefore incorrect.

Also, the postulated form of the coupling between degrees and weights is quite arbitrary, as well as its dependence on the parameter alpha.

The quality of the presentation is good.

D) Appropriate use of statistics and treatment of uncertainties:

Although the paper does not make use of a sophisticated statistical treatment of the data, or of advanced statistical tests, the comparison between the model and the data appears generally correct and it supports the conclusions.

E) Conclusions:

The conclusions are sound, and the accordance between the model and the data is a nice result. However, the robustness of these results is questionable given the highly "ad hoc" nature of the model. What new general insight do we get about network formation?

The problem of modelling weights and topology in a realistic and parsimonious way remains fundamentally unsolved.

F) Suggested improvements:

The paper should be submitted to a much more specialized journal like Physical Review E. The paper would benefit from a more intuitive discussion of what the abstract notion of geometric embedding plausibly means for real-world networks.

G) References:

Credit to previous work is not appropriately given. A vast literature on network embedding in various manifolds (including hyperbolic ones and higher-dimensional ones) exists. See the many works by Tomaso Aste and Tiziana di Matteo as an example. Also the main works about spatially embedded networks (see review by Marc Barthelemy and references therein) are not adequately cited.

H) Clarity and context:

abstract, introduction and conclusions are clear and well written. However they overstate the generality of the model and should more fairly emphasize the specific assumptions made in the paper.

Reviewer #2 (Remarks to the Author):

The authors approach the problem of whether the edge weights in real networks emerge have a geometrical origin and whether this can be well captured by suitable hidden variables models.

In particular, the authors first provide some evidence of the geometric nature of the weights, by studying the distribution of weights for links that are embedded in triangles. They then build a class of embedded weighted networks that is able to reproduce a number of properties of the observed networks. Finally, they show that this class of networks manages to capture and reproduces the observed metrical properties of real networks by checking the triangle violation in an hyperbolic embedding, equivalent to the first one.

The topic addressed is interesting, timely and relevant for the journal's audience.

It builds on previous work by some of the authors on hidden variable models and the corresponding embeddings that showed that many sparse unweighted networks can be fruitfully embedded in hyperbolic spaces yielding novel effective strategies for navigation and link prediction.

The paper is very well written and the topic clearly explained. The references and abstract are appropriate and cover the right existing literature on the subject.

The structure of the paper is appropriate and the contribution is novel. Previous work on the same subject focused on the unweighted case, this contribution provides first evidence that the description of weights too can be cast in the same paradigm.

Although I am a bit dubious about the long term impact of the paper, I believe it will be of interest to others in the field. I've have a few main comments (listed below), but I think that, once these are met, the manuscript will be fit for publication in Nature Communications.

Main comments:

- All the analysed networks have been sparsified via disparity filter before analysis.
- How does the analysis generalise to the case of dense networks?
- Pushing this argument, one might wonder what it would happen for the case of complete weighted networks, e.g. similarity or correlation networks, where the degree is already fixed. For example, for Pearson correlation matrices, the matrix already yields a distance matrix; how different does the weighted embedding proposed here would come out in that case?
- What is the role of D ? In this paper the authors provide insights on the geometry of edge

weights, but it's not clear the geometry of which space. Most of the analysed networks already live in a number of different dimensions (2, 3 etc) but they already appear to be well described by a $D=1$ model. It seems thus that the geometrical nature of the weights refers to a different geometrical space than the original network's natural embedding space. So, does this geometry really carry information/meaning or is it just a very general and elegant way to produce hidden variable networks? Alternatively, what would going to higher D grant in terms of network description or degrees of freedom?

- What are the atypical features that impeded the embedding of the US airport network? Do they constitute a problem for the general theory?

Minor comments:

- "particularize" is really an awkward word, maybe something like "we focus/restrict to the $D=1$ " or equivalent would be nicer.

- There's a typo on the als page "weigths" instead of "weights"

- Same page "On perspective" -> "in perspective"

Reviewer #3 (Remarks to the Author):

A. Summary of the key results

The authors Allard et. al. study the relationship between edge weights and a latent-space hyperbolic geometry for empirical networks. The latent spaces of networks are inferred using topology alone. The authors develop a novel network-generative model which they fit to empirical networks. They find that edge weights can be jointly coupled to the network topology (i.e., node degrees) as well as the geometry of the latent-space embedding. They explore the connection between edge weights and geometry by studying the violation of the triangle inequality for triangles in the network.

B. Originality and interest: if not novel, please give references

Understanding the origin of weights in weighted networks is a central topic in network science. This research provides the first step toward modeling weighted networks using latent-space, hyperbolic embeddings and is indeed an important contribution that should be published in some form. Publication in Nature Communications, however, requires a substantial advancement, and I believe the paper falls short in this regard. In particular, hyperbolic geometry is inferred from network topology, and since weights are known to depend on topology, it is somewhat unsurprising that there is a connection between the weights and geometry. For example, it has already been established that weights are larger for edges between larger degrees [1] (i.e., popularity) as well as for edges that join nodes with overlapping neighborhoods [19] (i.e., similarity). It is unclear whether or not the hyperbolic geometry modeling approach provides further insight than what is possible by studying the dependence of weights on node degrees and triangle participation (i.e., neighborhood overlap). (I note that both [1] and [19] are already cited in the paper, but the authors do not clearly discuss their connection to the geometric notions of 'popularity' and 'similarity'.)

C. Data & methodology: validity of approach, quality of data, quality of presentation

The methodology for hyperbolic space embeddings is state-of-the-art in the field of network science. Their model is indeed the state-of-the-art for modeling weighted networks in hyperbolic spaces.

D. Appropriate use of statistics and treatment of uncertainties

The article uses appropriate statistics, although it would be helpful to provide further details about their methods for inference.

E. Conclusions: robustness, validity, reliability

By modeling the coupling between weights, node degrees, and geometry, the authors provide a framework to deeply study these relationships. This is indeed an important contribution that justifies publication in some form. However, outside of observing, model fitting, and measuring the extent of these relations, very little other scientific insight is provided. That is, it is not clear if or how a relationship between weights and geometry will have an impact on any application.

F. Suggested improvements: experiments, data for possible revision

Main areas of improvement:

1. Section II studies the relationship between weights and triangles. Triangles and clustering reflect geometry due to the triangle inequality, however triangles are an indirect consequence of geometry. For examples, the number of triangles in which an edge is involved (that is, its multiplicity m) also depends on the nodes' degrees (i.e., topology). For example, in configuration models the multiplicity m grows with k_{ik_j} , since $(k_i-1)(k_j-1)$ gives number of possible triangles and edges are created at random. Given the focus on triangles both in section II and the violation of triangle inequality, the paper needs a much more detailed/systematic exploration and discussion of the relationship between triangles and geometry. This relation is currently vague, and citing the triangle inequality does not provide quantitative evidence of their connection.

2. In contrast to triangles, edge length is a direct measurement of geometry and may provide a more straightforward description for how weights and geometry are coupled. That is, are weights larger for shorter edges? Studying edge lengths may also help address comment A (the origin of clustering), since it would be helpful to understand if triangles primarily exist between node triples (i,j,k) that are nearby in the metric space, and if so, do they primarily involve nodes with small $\Delta \theta$ or nodes with large degrees.

3. It may also be informative to study the way in which triangle inequalities are violated. For example, is the inequality first violated for triangles involving nearby nodes or those that involve distant nodes? Is the inequality first violated for triangles involving hubs or those that do not involve hubs.

Minor issues:

4. abstract line 3: The authors do not 'prove' their model to be the "most" general and versatile model.

5. Sec. II - for many networks, multiplicity m and k_{ik_j} are highly correlated, implying that sampling of biased on m is similar to sampling with a bias on k_{ik_j} . It is worth noting how normalization according to k_{ik_j} overcomes this bias.

6. Sec. II - Does this normalization help address the goal of discerning the dependence of edge weights on $\Delta \theta$ versus k_i and k_j ?

7. Sec. III, Eq. 2 - secondary hidden parameters σ_i and σ_j are defined for edge

weights, however, it is later assumed that $\sigma_j = a k_j^\eta$. Why define them at all?

8. Sec. III - "given second moment $\langle k_i k_j \rangle < \epsilon^2$ " How is this chosen? Is it independent of k_i , k_j , and α ?

9. Sec. III - The statement "All the theoretical predictions are confirmed in Supplementary Figure 1." should be made more precise. i.e., what theoretical predictions? Scaling results?

10. Fig. 3 - The authors need to give a complete explanation of "atypical topological features".

11. Fig. 3 - Why do some TIV curves increase when $\alpha \sim 1$?

12. Discussion under Fig. 4 - "weigths" -> "weights"

G. References: appropriate credit to previous work?

The authors do a good job of citing previous research.

H. Clarity and context: lucidity of abstract/summary, appropriateness of abstract, introduction and conclusions

It may be helpful to discuss the triangle inequality and clustering in the abstract/intro given that it is a central topic of the paper. Also, I found the intro/abstract to not clearly identify new scientific insights allowed by the new model.

I. Summary

This research is an important and exciting area of network science, and the work is very high quality - both in philosophy and execution. However, I find the current paper to be lacking the "wow" factor that would justify publication in Nature Communications. The authors have made an interesting observation and developed a state-of-the-art model for it, but they have not illustrated this observation to have important consequences or provide useful insights. Moreover, I believe the "geometric nature" of weights to be under explored (see comments 1-2). For these reasons, I believe this work to be better suited for another journal.

1. Replies to the comments of Reviewer #1

We are glad to see that the reviewer judge that our “*approach is valid*”, that our “*conclusions are sound, and the accordance between the model and the data is a nice result*”, and that the “*abstract, introduction and conclusions are clear and well written*”. We thank the reviewer for her/his kind words. However, the reviewer made several remarks to which we would like to respond. Her/his comments are summarised in the following points.

1.A *“The main original contribution of this paper is a very specific model of geometrically embedded networks, where the weight structure is induced by the embedding.”*

We would like to stress that, in our opinion, the main contribution of our work is the empirical observation that weights in complex networks are influenced in a non-trivial way by some underlying metric structure. As far as we know, this is a novel and important result that extends the hidden/latent geometry paradigm to weighted complex networks. Of course, such empirical result claims for a modelling that would take it into account. This is the reason why we introduce our model, which is able to reproduce the coupling with the metric space in a very simple and elegant way. As we discuss in detail in the next point, our model has the ability to discriminate between purely local properties related to the degree and strength of nodes and the coupling with the metric space, thus giving us the ability to measure such coupling in real systems.

1.B *The approach is valid in itself, but the claim of being the most flexible and appropriate one is unjustified. The key assumption of the model, eq.(2), is a highly arbitrary choice. It relies on the assumption that there are two hidden variables per node, controlling for the expected degree and strength of that node respectively. Although the authors show that eq.(2) is the only one ensuring that the expected strength coincides with the associated hidden variable, this is an arbitrary and questionable criterion. Hidden variables are introduced precisely in order to ‘explain’ the expected properties (like strengths and degrees) via a hidden or underlying quantity which influences such properties, usually in a highly nonlinear way. Requiring the hidden variable to coincide with the expected value of the property it is controlling for is an unnecessary restriction. The claim of generality is therefore incorrect. Also, the postulated form of the coupling between degrees and weights is quite arbitrary, as well as its dependence on the parameter alpha.*

Notice that hidden variables can always be redefined such that they represent the property of interest, in our case nodes’ degrees and strengths. That being said, the model that we introduce in Eqs. (1) and (2) guaranties that we can fix the local properties of the nodes, that is, their joint degree-strength distribution similar to the one of a real network under study and, simultaneously, change in an independent manner the coupling between the weights and the metric space. This critical property is the one allowing us to gauge the effect of the metric space in real systems. This very same feature is also present in our different models of networks embedded in hidden (hyperbolic) spaces. There, too, we can fix the degree distribution and modify the coupling between topology and metric space so that different levels of clustering arise. This has been widely acknowledged by the community of network scientists and accepted as a new paradigm to describe and characterize complex networks. Our work here goes in the same direction and we are convinced that it will become the standard model for weighted networks embedded in metric spaces in the near future.

As for the particular form of Eq. (2) and the introduction of parameter α , here we follow a long standing tradition of “gravity models” in the Social Sciences, and in particular in Economics where the interaction between two countries is postulated to be proportional to the product of their “masses” (a measure of the importance of countries’ economies) and inversely proportional to their (geographic) distance. Eq. (2) is a novel generalisation of this concept to the case of weighted networks. In this case, the role of a given node’s “mass” must be played by the factor

$$\frac{\sigma}{\kappa^{1-\alpha/D}}$$

ensuring that, once the network has been assembled, that particular node has expected degree and strength κ and σ , respectively. Far from being arbitrary, the choice for this functional form of the “mass” and the identification of the hidden variables with the expected degrees and strengths ensure that, as a first but accurate approximation, we can use observed degrees and strengths in real networks as proxies for the unknown hidden variables.

Finally, our claim about the generality and versatility of our model is supported by the following properties:

1. Our model can fix in an arbitrary way the joint distribution $\rho(\kappa, \sigma)$, thus allowing to control the degree and strength distributions and any possible form of correlations (positive or negative) among degree and strength. In particular, $\rho(\kappa, \sigma)$ can take the form of correlations observed in real networks.
2. With strength and degree fixed, it can tune independently the coupling between weights and metric space through the parameter α and so reproduce the triangle inequality violation curves of real networks, as shown in Fig. 3.
3. The model can adjust the level of noise in the system through the parameter $\langle \epsilon^2 \rangle$. While noise is always present in real systems, it is usually not even considered in other models of weighted networks.
4. The model reproduces very well many other properties of real networks: degree-degree correlations, degree-dependent clustering coefficient, betweenness centrality (see for instance the supplementary information of Ref. [1]), global weight distribution, disparity measure, etc.

To the best of our knowledge, none of the models proposed so far in the literature satisfies all these characteristics simultaneously. In the new version of the manuscript, we have replaced the sentence “we introduce the most general and versatile class of weighted networks...” by “we introduce a very general and versatile class of weighted networks...”

1.C “The problem of modelling both the weights and topology of real networks is not new, but more than a decade old.”

We agree with the referee. Indeed, this is not a new problem. However, this fact does not make the problem uninteresting, but rather, it makes it more challenging (see for instance a couple of related works recently published in Nature Physics [2] and Nature Communications [3]). We would like to notice that despite the elapsed ten years, no much progress has been made in this field; the reason being the inherent difficulty of the problem. In our paper, we make a quite big step forward that, we are certain, will stimulate further research in this direction. On the one hand, the introduction of genuine gravity laws in networks will be of interest to fields where such laws can only be applied to fully connected structures. This opens a new line of theoretic research on the coupling between topology, weighted structure, and geometry in complex networks. On the other hand, our work opens the possibility to use information encoded in the weights of the links to find more accurate embeddings of real networks. We can then use these improved embeddings to detect network communities, missing links and, for the first time, give estimates of the weights of such missing links. We have modified the discussion section of our paper to emphasise these future research directions.

1.D “What new general insight do we get about network formation? The problem of modelling weights and topology in a realistic and parsimonious way remains fundamentally unsolved.”

In our opinion, the dependence of the properties of networks, both at the topological and weighted levels, on hidden/latent metric spaces which encode in a geometric distance all the factors that affect the propensity of nodes to establish connections and to supply them with a certain intensity is a new insight about network formation that we consider very interesting and important and that should be taken into account in any future research on this topic. For the first time, the gravity model can be used to explain both the formation of links and weights in networks. On top of that, we provide a general model that, for the first time, accurately reproduces many properties observed in real weighted networks from various origins. Incidentally, our model unveils that the coupling of the topology and the weights with the underlying metric space are in some cases uncorrelated, which in turn suggests that the formation of connections and the assignment of their magnitude can be ruled by different processes. We agree with the reviewer that many questions about the formation of real weighted networks remain open, and we believe that our contribution is substantial enough to encourage other researchers to join the endeavour.

1.E “What the abstract notion of geometric embedding plausibly means for real-world networks?”

This is indeed an excellent and difficult question that we have been trying to answer since we published our first work on the subject [4]. In that work, we found that the properties of the degree-dependent clustering

coefficient of some real complex networks are compatible with the existence of a hidden metric space ruling the probability of existence of links between nodes. Interestingly, we also found that the Internet fits particularly well within this new paradigm and its inferred embedding in the hyperbolic plane (see Ref. [1]) has excellent routing properties in this space. We have also developed both static and growing models (like the one in Ref. [5]) showing that our models are able to reproduce the topologies of real complex networks extremely well and, at the same time, are mathematically tractable as interactions are still pairwise.

The nature of such hidden metric spaces is, however, not totally clear. In the case of social structures for instance, even though it is difficult to quantify, we, as individuals, are able to tell whether one person is close or far from us and, in many cases, we establish social relationships based on this perception. In the social sciences, this concept is called homophily and is responsible for the assortative character of social networks. In the case of economic systems, like countries, the metric distance can be an effective space combining geographic, cultural, historical, and political distances (see for instance Ref. [6]). In the case of the Internet, it is probably a combination of geography and commercial agreements between the different actors at play.

Nevertheless, beyond the philosophical discussion about the origin of such spaces, the concept of a hidden metric space can also be seen as a mathematical tool that can be leveraged to generate realistic networks. For instance, it is the only framework that allows to generate strong clustering based on pairwise interactions only (due to the triangle inequality of the metric space). Similarly, our manuscript demonstrates how the metric space allows to realistically and directly assign weights to links given a fixed degree-strength sequence (i.e., without relying on iterative methods), which is a notoriously difficult problem.

1.F *“Credit to previous work is not appropriately given. A vast literature on network embedding in various manifolds (including hyperbolic ones and higher-dimensional ones) exists.”*

We thank the reviewer for pointing out potential interesting references, and we now acknowledge the work of Aste and Di Matteo as well as of Barthelemy (whose work was already acknowledged via other references). However, while the work of Aste and di Matteo embed complex networks into hyperbolic manifolds to characterise and filter them (see for instance [7-8]), the information encoded in the distance between nodes remains unexploited and, as such, their many contributions are only weakly related to our work (and most of the work already cited in the manuscript).

2. Replies to the comments of Reviewer #2

We are delighted that the reviewer finds that the *“topic addressed is interesting, timely and relevant for the journal’s audience”*, that the *“paper is very well written and the topic clearly explained”*, that our *“contribution is novel”*, and that our *“manuscript will be fit for publication in Nature Communications”* once a few minor comments have been addressed. The latter are addressed below.

2.A *“How does the analysis generalise to the case of dense networks? Pushing this argument, one might wonder what it would happen for the case of complete weighted networks, e.g. similarity or correlation networks, where the degree is already fixed. For example, for Pearson correlation matrices, the matrix already yields a distance matrix; how different does the weighted embedding proposed here would come out in that case?”*

The reviewer raises a good point. The analysis of the model presented in the Supplementary Information shows that the networks generated by our model are sparse in the limit $N \rightarrow \infty$. This result assumes, however, that the density δ , the average expected degree $\langle \kappa \rangle$ and the integral I_1 are all bounded. Reference [9] studies a special case in which I_1 is not bounded but shows that the average degree of the network scales sub-linearly with the number of nodes. It is not clear, however, how the model behaves in general in the case of dense or complete networks, which would require a different model to generate the topology. In fact, current ongoing research is looking into a variation of our model and its application to correlation matrices.

2.B *“What is the role of D ? In this paper the authors provide insights on the geometry of edge weights,*

but it's not clear the geometry of which space. Most of the analysed networks already live in a number of different dimensions (2, 3 etc) but they already appear to be well described by a $D=1$ model. It seems thus that the geometrical nature of the weights refers to a different geometrical space than the original network's natural embedding space. So, does this geometry really carry information/meaning or is it just a very general and elegant way to produce hidden variable networks? Alternatively, what would going to higher D grant in terms of network description or degrees of freedom?"

The reviewer is right *"that the geometrical nature of the weights refers to a different geometrical space than the original network's natural embedding space"*. In fact, one of the networks for which our model works best is the metabolic network, which does not have a natural embedding space. The hidden metric space used in our framework is an abstract space in which the distance between nodes encodes the likelihood for them to be connected. Notice, however, that not only the geometrical distance (i.e., the arc length in the case of the circle \mathbb{S}^1) influences the likelihood of nodes to be connected, but also the product of their respective expected degrees [see Eq. (1) and (3)]. In other words, two hubs are effectively closer than two low-degree nodes even if both pairs are separated by the same arc length.

This is particularly important in the case of heterogeneous (scale-free) networks because this makes the particular dimension of the metric space, D , not so relevant. The reason is, as we show in Ref. [9], that our model can be mapped into a purely geometric random graph in the $(D + 1)$ -dimensional hyperbolic space \mathbb{H}^{D+1} . In such space, the volume of a ball of radius r grows as $V \sim e^{Dr}$ and, thus, the value of D only changes the pre-factor of the exponential growth law of the ball but not the fact that it grows exponentially. This implies that, even if the original metric space is not one-dimensional, its embedding in the two-dimensional hyperbolic plane is very good. In other words, \mathbb{H}^2 already has *enough space* to fit any network without violating the triangle inequality. In our paper we chose to leave the parameter D free for the sake of presenting the most general model possible. However, when it comes to studying real networks, and given the considerations above, we chose $D = 1$ as it simplifies enormously the analytic and computational treatment. Of course, one could however argue that going to higher dimensions—and therefore having more degrees of freedom in the geometrical space—could be useful in the context of communities [10], for instance. While this is perfectly possible from a theoretical point of view, the inverse problem of finding embeddings of real networks would become computationally infeasible.

2.C "What are the atypical features that impeded the embedding of the US airport network? Do they constitute a problem for the general theory?"

The atypical features mentioned in the manuscript refer to power-law degree distribution with an exponent below 2 in the case of the U.S. airports network and a short-range repulsion effect in the connection probability for the commute network (i.e., people rarely commute from one suburb to another but rather commute from one suburb to the major city in the area). This does not affect the general theory described in the manuscript but rather prevent the state-of-the-art embedding algorithms to provide us with an embedding of the two networks. We are currently working on a generalisation of a class of embedding algorithms that would allow the embedding of such networks.

2.D Minor comments

We thank the reviewer for pointing our these typos which have been corrected accordingly.

2.E "The references and abstract are appropriate and cover the right existing literature on the subject."

We thank the reviewer.

3. Replies to the comments of Reviewer #3

We acknowledge that the reviewer deems that our contribution tackles *"an important and exciting area of network science"* by providing *"the first step toward modelling weighted networks latent-space, hyperbolic embeddings"*, that our model is *"state-of-the-art for modelling weighted networks in hyperbolic space"*, that our *"work is very high quality—both in philosophy and execution"* and that it is *"indeed an important*

contribution that should be published in some form". We thank the reviewer for her/his kind words. As for reviewer #1, the reviewer expresses some doubts as to whether our contribution is fit to be published in Nature Communications. We respond point by point to her/his comments in hope to convince her/him of that our contribution is worthy of the high standards of this journal.

3.A "In particular, hyperbolic geometry is inferred from network topology, and since weights are known to depend on topology, it is somewhat unsurprising that there is a connection between the weights and geometry. For example, it has already been established that weights are larger for edges between larger degrees [1](i.e., popularity) as well as for edges that join nodes with overlapping neighbourhoods [19] (i.e., similarity). It is unclear whether or not the hyperbolic geometry modelling approach provides further insight than what is possible by studying the dependence of weights on node degrees and triangle participation (i.e., neighbourhood overlap). (I note that both [1] and [19] are already cited in the paper, but the authors do not clearly discuss their connection to the geometric notions of "popularity" and "similarity.")"

Please notice that to perform the empirical analysis in Fig. 1, we do not infer the hyperbolic geometry from network topology (which we do in the final part of the paper for self-consistency of the analysis). Instead, we study how normalised weights are distributed over the edges of the network. We agree with the reviewer that weights in complex networks depend on the topology (see Ref. [11]) and this is precisely why we considered normalised weights in Section II of the manuscript (see also our answer to comment 3.1). By normalising the weights by the average value $\bar{\omega}(kk')$, we factorised out the dependency on the topology, leaving weights that seemingly randomly fluctuate around 1. However, as shown in Fig. 1, these fluctuations are not uniform, as we see that links involved in triangles tend to have larger normalised weights than the average link. Since triangles are a reflection of the triangle inequality in the underlying metric space, we expect nodes forming triangles to be close to one another. Thus, the higher average normalised weight observed on triangles strongly suggests a metric nature of weights, which is not a trivial consequence of the relation between weights and topology. Notice also that our theoretical model provides a counterexample of the referee's observation that *"since weights are known to depend on topology, it is somewhat unsurprising that there is a connection between the weights and geometry"*. Indeed, by changing the parameter α , we can generate networks with an arbitrary coupling between weights and metric space (even zero coupling) even though they share the very same network topology and correlations between strength and degree. However, we found that the weight distribution and the disparity were well reproduced only with the value of α found using the test of the triangle inequality.

As for the results in Refs. [1] and [19] (in the old version of the manuscript), it is indeed well known and accepted that weights are higher between nodes of high degrees. In [19], the authors found a positive correlation between weights and link clustering (or neighbourhood overlap). However, since it is also true that link clustering is typically correlated with the degrees of the endpoint nodes, the correlation between absolute weights and degrees is also expected, which prevents from a direct observation of metric properties in the weights. In our work, we filter out such induced correlations by normalising the weights so that genuine correlations with the metric space can be detected. In the revised version of the paper, we have included a discussion to clarify this point.

3.B "The article uses appropriate statistics, although it would be helpful to provide further details about their methods for inference."

In section II of the Supplementary Information file, we provide a detailed explanation of the statistical method we have developed to measure parameters α and $\langle \epsilon^2 \rangle$. The embedding methods of the unweighted versions of the networks is fully described in our previous publication [1].

3.C "By modelling the coupling between weights, node degrees, and geometry, the authors provide a framework to deeply study these relationships. This is indeed an important contribution that justifies publication in some form. However, outside of observing, model fitting, and measuring the extent of these relations, very little other scientific insight is provided. That is, it is not clear if or how a relationship between weights and geometry will have an impact on any application. "

With hindsight, we agree with the reviewer that we may have lacked in explaining clearly the implications of our work for the understanding of real weighted networks, which we believe are remarkable. For instance, our equations can be understood as the new generation of gravity laws applicable to very different domains,

including Biology, Information and Communication Technologies, and Social Systems. Current gravity laws are prescribed to the Social Sciences and predict successfully the volume of flows between elements but cannot explain the observed topology of the interactions among them. Our contribution overcomes this limitation and offers for the first time a gravity model that can reproduce both the existence and the intensity of interactions. This opens a new line of theoretic research on the coupling between topology, weighted structure, and geometry in complex networks. On the other hand, our work opens the possibility to use information encoded in the weights of the links to find more accurate embeddings of real networks. We can then use these improved embeddings to detect network communities, missing links and, for the first time, give estimates of the weights of such missing links, and to implement navigation and searching protocols, such as greedy routing, which take into account not only the existence of connections but also their intensity. We have modified the discussion section of our paper to emphasise these future research directions.

3.D “Section II studies the relationship between weights and triangles. Triangles and clustering reflect geometry due to the triangle inequality, however triangles are an indirect consequence of geometry. For examples, the number of triangles in which an edge is involved (that is, its multiplicity m) also depends on the nodes’ degrees (i.e., topology). For example, in configuration models the multiplicity m grows with $k_i k_j$, since $(k_i - 1)(k_j - 1)$ gives number of possible triangles and edges are created at random. Given the focus on triangles both in section II and the violation of triangle inequality, the paper needs a much more detailed/systematic exploration and discussion of the relationship between triangles and geometry. This relation is currently vague, and citing the triangle inequality does not provide quantitative evidence of their connection.”

We would like to notice that in the configuration model the link multiplicity vanishes in the thermodynamic limit and, therefore, such model is not a good candidate to have an underlying geometry. In networks with finite clustering, like in our model, there is typically a non-trivial relation between $m_{kk'}$ and k and k' , although this relation is, in general, difficult to calculate. In general, the problem of measuring the metricity of network topologies is an extremely difficult problem that requires a research program on its own. Nevertheless, very useful information can be obtained from the properties of the clustering in the network. The relation between clustering and geometry has been analysed in detail in our previous publications. In the current paper, we take the relation clustering/metric space for granted and we focus on the relation between weights and geometry. It is true that in our geometric models clustering is a by-product of the metric space (and so of the triangle inequality), which is, by the way, very convenient from a mathematical point of view, as it induces effective three body interactions from pairwise ones. However, in our first publication on this topic (Ref. [4]) we found that the properties of the degree-dependent clustering coefficient of some real complex networks are compatible with the existence of a hidden metric space ruling the probability of existence of links between nodes. Interestingly, we also found that the Internet fits particularly well within this new paradigm and its inferred embedding in the hyperbolic plane (see Ref. [1]) has excellent routing properties in this space. We have also developed both static and growing models (like the one in Ref. [5]) showing that our models are able to reproduce the topologies of real complex networks extremely well.

3.E “In contrast to triangles, edge length is a direct measurement of geometry and may provide a more straightforward description for how weights and geometry are coupled. That is, are weights larger for shorter edges? Studying edge lengths may also help address comment A (the origin of clustering), since it would be helpful to understand if triangles primarily exist between node triples (i, j, k) that are nearby in the metric space, and if so, do they primarily involve nodes with small $\Delta\theta$ or nodes with large degrees.”

We thank the reviewer for pointing out a concept that may not have been sufficiently well explained in our manuscript. The distance between nodes in the metric space does not correspond to the actual geographical distance between, say, airports in the US airports network. Rather, it is an abstract distance that quantifies the likelihood of interactions between nodes. Consequently, a direct measurement is not available and this is why we turned to triangles—as a reflection of the triangle inequality in the metric space—as a proxy to estimate qualitatively the distance between nodes (i.e., close or distant). To answer directly to the reviewer’s first question, Eq. (2) in the manuscript stipulates that the weight between connected nodes should decrease with increasing distance between them in the hidden metric space.

Triangles between node triples (i, j, k) exist with probability $p(\chi_{ij})p(\chi_{jk})p(\chi_{ik})$, a quantity that essentially

Figure 1: Average degree of nodes in triangles that violate the triangle inequality ($\langle k \rangle_{TIV}$) and in all triangles ($\langle k \rangle_{Triangle}$) for the networks considered in the main text. The average is performed by sampling over triangles which implies that the degree of a node is weighted by the number of triangles to which it participates (as in Fig. 1 of the main text). The dashed line shows the fraction of triangles that violate the triangle inequality when using the inferred value α_{real} .

depends on the ratios $\frac{\Delta\theta_{ij}}{\kappa_i\kappa_j}$, $\frac{\Delta\theta_{jk}}{\kappa_j\kappa_k}$ and $\frac{\Delta\theta_{ik}}{\kappa_i\kappa_k}$. In other words, triples with low $\Delta\theta$ s will likely form a triangle regardless of their expected degrees just as triples with high expected degrees will likely form a triangle regardless of their position on the circle S^1 . Similarly, the probability for triangles involving two hubs and one low degree node will not strongly depend on the relative position of the nodes on the circle. However, triples with one hub and two low degree nodes will typically not form triangles unless the two low degree nodes are very close along the circle. In fact, this effect contributes greatly to explain why the degree-dependent clustering $\bar{c}(k)$ is a decreasing function of k for all networks considered in the manuscript (see Supplementary Figures 6–11).

3.F “It may also be informative to study the way in which triangle inequalities are violated. For example, is the inequality first violated for triangles involving nearby nodes or those that involve distant nodes? Is the inequality first violated for triangles involving hubs or those that do not involve hubs.”

This is a very interesting question. The triangle inequality violation curves are used to find the parameter α_{real} of a given network. Once this value is found, the violation of the triangle inequality depends essentially on the level of noise in the system ($\langle \epsilon^2 \rangle$) through the term in the right hand side of Eq. (7). To a lesser extent, the violation may also be due to the fact that the hidden variables κ and σ are approximated by the actual degree and the strength, respectively, of nodes. For most of the analysed real networks, the percentage of violations is very small (of the order of few percent) whereas in the case of the cargo ships network it is close to 20%, due to the high level of noise present in the system. In short, our model predicts that there should not be any dependence on the degree in the nodes belonging to triangles that violate the triangle inequality. To test this prediction, we have measured explicitly the average degree of such nodes as compared to the average degree of nodes in all triangles (see Fig. 1). In many cases the average degree is very similar, thus confirming our prediction. The largest discrepancy is found in the metabolic network. However, notice that this network has a very small percentage of violations, which makes it more prone to statistical fluctuations. We have added a discussion in the new version of the Supplementary Information to clarify this point.

3.G “Abstract line 3: The authors do not “prove” their model to be the “most” general and versatile model.”

Our claim about the generality and versatility of our model is supported by the following properties:

1. Our model can fix in an arbitrary way the joint distribution $\rho(\kappa, \sigma)$, thus allowing to control the degree and strength distributions and any possible form of correlations (positive or negative) among degree and strength. In particular, $\rho(\kappa, \sigma)$ can take the form of correlations observed in real networks.
2. With strength and degree fixed, it can tune the coupling between weights and metric space through the parameter α and so reproduce the triangle inequality violation curves of real networks, as shown in Fig. 3.
3. The model can adjust the level of noise in the system through the parameter $\langle \epsilon^2 \rangle$. While noise is always present in real systems, it is usually not even considered in other models of weighted networks.
4. The model reproduces very well many other properties of real networks, degree-degree correlations, degree-dependent clustering coefficient, betweenness centrality (see for instance the Supplementary Information of Ref. [1]), global weight distribution, disparity measure, etc.

To the best of our knowledge, none of the models proposed in the literature satisfies all these characteristics simultaneously. In the new version of the manuscript, we have replaced the sentence “we introduce the most general and versatile class of weighted networks...” by “we introduce a very general and versatile class of weighted networks...”

3.H *“Sec. II - for many networks, multiplicity m and $k_i k_j$ are highly correlated, implying that sampling of m is similar to sampling with a bias on $k_i k_j$. It is worth noting how normalisation according to $k_i k_j$ overcomes this bias. ”*

Notice also that since we measure weights normalised by the average weight $\bar{\omega}(kk')$, a biased sampling over m should be equivalent to a uniform sampling provided there is no metric space dependence on the weights. Any deviation indicates correlations between clustering and weights.

3.I *“Sec. II - Does this normalisation help address the goal of discerning the dependence of edge weights on $\Delta\theta$ versus k_i and k_j ?”*

Yes it does. It has been observed that the average weight of links whose end nodes have degrees k and k' scales as $\bar{\omega}(kk') \sim (kk')^\tau$ where $\tau = 0.5 \pm 0.1$ in the case of the international airports network (see Ref. [11]). However, we found in all our datasets that while the average weight does depend on the product kk' , this dependency cannot be summarised in a form as simple as the one proposed in Ref. [11]. For instance, two different scaling regimes could be observed in some datasets. Consequently, we decided to let the datasets speak by themselves by not imposing a specific analytical form for the dependency of the average weight over kk' and simply divide the weights by the average $\bar{\omega}(kk')$. By doing so, we removed the influence of the topology on the weights, which allows to unveil their metric origin.

3.J *“Sec. III, Eq. 2 - secondary hidden parameters σ_i and σ_j are defined for edge weights, however, it is later assumed that $\sigma_j = ak_j^\eta$. Why define them at all?”*

We agree with the reviewer that the specific application of our model in the manuscript does not require a second hidden variable σ since it is linked to the first hidden variable κ via a deterministic relation $\sigma = a\kappa^\eta$. However, as demonstrated in the Supplementary Information, the second hidden variables, σ , correspond to the expected strength of nodes regardless of the relation with the first hidden variable κ . In other words, our model is much more general and versatile and we consider that this feature is worth mentioning in the manuscript.

3.K *“Sec. III - “given second moment $\langle \epsilon^2 \rangle$ ” How is this chosen? Is it independent of k_i , k_j , and α ?”*

The second moment $\langle \epsilon^2 \rangle$ is a global parameter of the model and, as such, is independent of the degree of nodes. However, it is dependent in the coupling parameter α between the weights and the metric space. The details of how the value of $\langle \epsilon^2 \rangle$ is chosen is explained in detail in the Supplementary Information.

3.L *“Sec. III - The statement “All the theoretical predictions are confirmed in Supplementary Figure 1.” should be made more precise. i.e., what theoretical predictions? Scaling results?”*

Figure 2: α -dependent terms of the right hand side of Eq. (1) as a function of α for the real networks considered in the main text.

All the theoretical predictions are derived in the Supplementary Information and are summarised in Sec. III of the manuscript. We have modified the sentence the reviewer is referring to accordingly.

3.M “Fig. 3 - The authors need to give a complete explanation of “atypical topological features”.”

We refer the reviewer to our answer to the comment 2.C.

3.N “Fig. 3 - Why do some TIV curves increase when $\alpha \sim 1$?”

The increase of $TIV(\alpha)$ close to $\alpha = 1$ on Figs. 3a–b is expected and is in fact a consequence of Eq. (7) and of our choice of the probability of connection [i.e., Eq. (3)]. Indeed, substituting Supp. Eq. (23) in Eq. (7) in the main text and neglecting the noise term (whose mean value is close to zero) we obtain

$$\ln \left[\frac{\omega_{ij}\omega_{jk}}{\omega_{ik}} \left(\frac{\kappa_j}{\sigma_j} \right)^2 \right] \leq \frac{R}{2}\alpha + \ln \left(\sin \left[\frac{(1-\alpha)\pi}{\beta} \right] \right) + \ln \left(\frac{\beta}{2\pi\mu\langle\sigma\rangle} \right). \quad (1)$$

Figure 2 below shows the behaviour of α -dependent terms of the right hand side of Eq. (1) for the real networks considered in the main text. For low values of α , we see that the right hand side of Eq. (1) is an increasing function which implies that $TIV(\alpha)$ decreases with increasing α (i.e., it is *more and more difficult* to violate the triangle inequality as α increases). However, all curves reach a plateau at $\alpha \simeq 0.8$ after which they start to decrease. As expected, these plateaus correspond to the points where the $TIV(\alpha)$ start to increase (for some networks this increase is not visible due to the linear scale of the y axis). This discussion has been added to the Supplementary Information.

3.O “12. Discussion under Flg. 4 - “weights” -> “weights””

The typo has been corrected.

3.P “The authors do a good job of citing previous research.”

We thank the reviewer for this appreciation.

3.Q “It may be helpful to discuss the triangle inequality and clustering in the abstract/intro given that it is a central topic of the paper. Also, I found the intro/abstract to not clearly identify new scientific insights allowed by the new model.”

We thank the reviewer for the suggestion and we have mentioned the triangle inequality and emphasised more on the scientific insights in the abstract, in the introduction and in the discussion.

3.R “This research is an important and exciting area of network science, and the work is very high quality - both in philosophy and execution. However, I find the current paper to be lacking the “wow” factor that

would justify publication in Nature Communications. The authors have made an interesting observation and developed a state-of-the-art model for it, but they have not illustrated this observation to have important consequences or provide useful insights."

We thank the reviewer for her/his kind words about the quality of our work. However, we believe that a "wow" factor is a very subjective feeling, especially as other readers (like reviewer #2) might think differently. As mentioned in our answer to comment 3.C, we agree with the reviewer that we may have lacked in explaining clearly the implications of our work for the understanding of real weighted networks, which we believe are remarkable.

We would like to stress that, in our opinion, the main contribution of our work is the empirical observation that weights in complex networks are influenced in a non-trivial way by some underlying metric structure. As far as we know, this is a novel and important result that extends the hidden/latent geometry paradigm to weighted complex networks. Of course, such empirical result claims for a modelling that would take it into account. This is the reason why we introduce our model, which is able to reproduce the coupling with the metric space in a very simple and elegant way. Our model guarantees that we can fix the local properties of the nodes, that is, their joint degree-strength distribution similar to the one of a real network under study and, simultaneously, change in an independent manner the coupling of the weights with the metric space. This critical property is the one allowing us to gauge the effect of the metric space in real systems. This very same feature is also present in our different models of networks embedded in hidden (hyperbolic) spaces. There, too, we can fix the degree distribution and modify the coupling between topology and metric space so that different levels of clustering arise. This has been widely acknowledged by the community of network scientists and accepted as a new paradigm to describe and characterise complex networks. Our work here goes in the same direction and we are convinced that it will become the standard model for weighted networks embedded in metric spaces in the near future.

At the same time, our equations can be understood as the new generation of gravity models applicable to very different domains, including Biology, Information and Communication Technologies, and Social Systems. Current gravity laws are prescribed to the Social Sciences and predict successfully the volume of flows between elements but cannot explain the observed topology of the interactions among them. Our contribution overcomes this limitation and offers for the first time a gravity model that can reproduce both the existence and the intensity of interactions. This opens a new line of theoretic research on the coupling between topology, weighted structure, and geometry in complex networks. On the other hand, our work opens the possibility to use information encoded in the weights of the links to find more accurate embeddings of real networks. We can then use these improved embeddings to detect network communities, missing links and, for the first time, give estimates of the weights of such missing links, and to implement navigation and searching protocols, such as greedy routing, which take into account not only the existence of connections but also their intensity. We have modified the discussion section of our paper to emphasise these future research directions.

Bibliography

- [1] Boguñá, M., Papadopoulos, F., & Krioukov, D. (2010). Sustaining the Internet with hyperbolic mapping. *Nat. Commun.*, 1, 62. <http://doi.org/10.1038/ncomms1063>
- [2] Pajevic, S., & Plenz, D. (2012). The organization of strong links in complex networks. *Nature Phys.*, 8, 429–436. [doi:10.1038/nphys2257](http://doi.org/10.1038/nphys2257)
- [3] Grady, D., Thiemann, C., & Brockmann, D. (2012). Robust classification of salient links in complex networks. *Nat. Commun.*, 3, 864. [doi:10.1038/ncomms1847](http://doi.org/10.1038/ncomms1847)
- [4] Serrano, M. Á., Krioukov, D., & Boguñá, M. (2008). Self-similarity of complex networks and hidden metric spaces. *Phys. Rev. Lett.*, 100, 78701. <http://doi.org/10.1103/PhysRevLett.100.078701>
- [5] Papadopoulos, F., Kitsak, M., Serrano, M. Á., Boguñá, M., & Krioukov, D. (2012). Popularity versus similarity in growing networks. *Nature*, 489, 537–540. <http://doi.org/10.1038/nature11459>

- [6] Ghemawat, P. (2001). Distance Still Matters: The Hard Reality of Global Expansion. *Harvard Business Review* 79, 137-147. hbr.org/2001/09/distance-still-matters-the-hard-reality-of-global-expansion
- [7] Di Matteo, T., & Aste, T. (2005). Extracting the correlation structure by means of planar embedding. In *Proc. SPIE* (Vol. 6039, p. 60390P–60390P–10). doi:10.1117/12.637543
- [8] Tumminello, M., Aste, T., Matteo, T. Di, & Mantegna, R. N. (2005). A tool for filtering information in complex systems. *Proc. Natl Acad. Sci. USA*, 102, 10421–10426. doi:10.1073/pnas.0500298102
- [9] Krioukov, D., Papadopoulos, F., Kitsak, M., Vahdat, A., & Boguñá, M. (2010). Hyperbolic geometry of complex networks. *Phys. Rev. E*, 82, 036106. doi:10.1103/PhysRevE.82.036106
- [10] Zuev, K., Boguñá, M., Bianconi, G., & Krioukov, D. (2015). Emergence of Soft Communities from Geometric Preferential Attachment. *Scientific Reports*, 5, 9421. doi:10.1038/srep09421
- [11] Barrat, A., Barthélemy, M., Pastor-Satorras, R., & Vespignani, A. (2004). The architecture of complex weighted networks. *Proc. Natl Acad. Sci. USA*, 101, 3747–52. doi:10.1073/pnas.0400087101

Reviewers' comments:

Reviewer #1 (Remarks to the Author):

After having read the new version of the manuscript and the authors' responses to the referees' remarks, I remain skeptical about the significance of these results and their suitability for Nature Communications. As a consequence, I still do not recommend publication of this manuscript.

- In their reply 1.E, the authors write "...the concept of a hidden metric space can also be seen as a mathematical tool that can be leveraged to generate realistic networks. For instance, it is the only framework that allows to generate strong clustering based on pairwise interactions only (due to the triangle inequality of the metric space)".

This is not true: even random graphs with given degrees can have a large clustering. This is not often recognised, but dates back to Park and Newman's PRE paper "Origin of degree correlations in Internet and other networks". The reason why this result is overlooked is the widespread use of the approximation that factorizes the connection probability into the product of the end-point degrees. As originally showed by Maslov, Sneppen and collaborators, this approximation is inconsistent with the large value of the maximum degree in real-world networks. If realistic degree sequences are to be replicated, one needs to go beyond the naive factorized approximation. The resulting probability of connection is highly nonlinear (it has a Fermi-function shape) and was derived by Park and Newman in the paper above and in many subsequent papers. This probability function is the correct one for a network with broad degree distribution and generates a high level of clustering (often matching perfectly the empirical clustering), even if it only accounts for local (degree) properties of nodes, without resorting to any metric pairwise property.

- Therefore the apparent need to introduce metric spaces to replicate high clustering might be merely an artefact of the (incorrect) approximation of the connection probability. Note that, even if it is often said that the factorized probability works well for "sparse networks", this is actually incorrect: a factorized probability does generate sparse networks, but these networks are however unrealistic in terms of their maximum degree. In other words, real-world networks, although sparse in most cases, are incompatible with the factorized approximation. Compensating this unrealistic approximation with the introduction of a metric space in order to retrieve an otherwise unexplained large clustering is scientifically incorrect and misleading. The large clustering (or at least a generous portion of it) would more parsimoniously be explained by using only local properties (e.g. degrees), along with the correct nonlinear connection probability accounting for them.

- Even though such a factorized approximation is never introduced explicitly as a building block of the model described in this paper, an equivalent problem is present here in terms of the expected weight being linearly dependent on the expected strengths. Indeed, as a consequence of this approximation, in the model proposed the expected strengths turn out to be proportional to the corresponding hidden variables, apparently justifying the claim that their model can account for any (joint) degree and strength distribution. Again, this claim of generality is not founded and the resulting metric "patterns" might be an artefact compensating for the factorised choice of the expected weights.

- Additionally, since the authors want to decouple local node effects (degrees and strengths) from the (postulated) metric properties, it is not clear why they preliminary filter most of their networks with the "disparity filter" (by the way, why don't they do this for all networks? In the SI they say that some networks have been filtered this way, and others not, without explanation). By using this filter, the local effects should in principle vanish, so they should be left with "residual networks" where local node properties are no longer relevant and should not be further controlled for. So why are they applying their model to these filtered networks? This procedure is unclear to me. In any case, it raises the doubt whether the empirical "patterns" that are documented here are actually properties of how the disparity filter operates, rather than properties of the data

themselves.

- By the way, the disparity filter assumes that the total strength s of a node is uniformly randomly broken up into the weights of the k edges coming out of a node, irrespective of the degrees at the other endpoint of these edges. This again appears to contrast the well-known fact, used also elsewhere in this paper, that connection probabilities should depend on the degrees at both endpoints of an edge. So here I see some inconsistency in the way data are analysed.

- Finally, it is not true that this is the first "generation of gravity models" assuming that also the probability of connections should be a gravity-like function. There is vast literature about the so-called zero-inflated gravity models which do have a similar dependence of link probabilities on the gravity equation, thus replicating the observed network density (see for instance the published papers by Fagiolo (<http://arxiv.org/abs/0908.2086>) and Fagiolo and Duenas (<https://arxiv.org/abs/1112.2867>) and references therein.

In conclusion, I still believe that this papers does not introduce a really new and general paradigm to explain the origin of weights in real networks. It might be forcing the use of metric spaces to compensate for some implicit proportionality assumption (for sure it is partially doing so), it might be partially looking at spurious patterns created by the filtering method used, and it is not the first/only model that has been proposed to understand the empirical weights in weighted networks.

Reviewer #2 decided to provide confidential remarks to the editor only. In them, they continue to praise the value of your work, and believes that your contribution deserves publication in Nature Communications. At the same time, they explain that in their view some of the criticisms of Reviewers #1 and #3 may be based on the natural difficulty of the language required to describe hyperbolic embeddings, and because of the objective difficulty in interpreting what an underlying and new hyperbolic metric structure is really telling us about the networks under study. And while they believe that you did already a very good job regarding the former point, in terms of explaining your method, they concede that regarding the latter a solid answer to the origin of the weights and direct interpretation of the uncovered hyperbolic structure has not been yet provided. Nonetheless, they remain positive towards the work in light of the potential to stimulate new work that it has.

Reviewer #3 (Remarks to the Author):

I have examined the authors' revised manuscript and their responses to my comments. Although several of my concerns have been adequately addressed, the authors did not directly address several of the main issues that I previously raised.

As I previously stated, my overall feeling is that the paper provides a nice contribution to this field and deserves publication in some form and at some venue. However, I cannot support publication in Nature Communications until the issues below are carefully addressed in the manuscript. That said, I now believe the manuscript to be sufficiently impactful to warrant publication in Nature Communications. My recommendation is now 'revise and resubmit.'

Previous concerns not adequately addressed:

(3.A). I believe the authors missed my main concern, which regards my previous statement 'It is unclear whether or not the hyperbolic geometry modeling approach provides further insight than what is possible by studying the dependence of weights on node degrees and triangle participation

(i.e., neighbourhood overlap).' I will further explain this concern.

Specifically, given the observations that node degree and triangle participation both influence edge weights, the simplest model would be one in which weights depend only on two types of variables: node degrees and triangle participation. My concern regards whether or not the complicated latent-geometry model satisfies the Occam's razor principle. I believe that it does, but given the complexity of their model, the authors should provide strong evidence and a clear discussion for why the hyperbolic-geometry model is superior to a simpler alternative.

I point out that a correlation between triangle participation and edge weight is widely believed, despite -- as identified by the authors -- some results in [19] are lacking evidence since they do not isolate the effect some of their experiments. I agree that the authors conduct a more principled experiment with Fig. 1, but the main message of Fig. 1 (i.e., triangles influence edge weights) is not a new idea. It is actually the focus of [19], which is a paper that includes more results than the single experiment upon which the authors improve.

The authors' novel claim with Fig. 1 is that a hidden geometry is the origin of this phenomenon. That is, the correlation between triangle participation and edge weight is (or can be) an artifact of a correlation between geometry and edge weight. I believe such a claim requires two types of support:

- (i) The hidden geometry model can account for the correlation between edge weight and triangle participation.
- (ii) The hidden geometry model provides a 'better' explanation versus a much simpler model in which edge weights only depend on node degrees and triangle participation.

(i) is strongly supported by their study of TIV curves. In my opinion, (ii) is insufficiently described in the paper. That is, the authors do not clearly explain why adopting a complicated latent-space model for edge weights is superior to a simpler alternative model in which one only takes into account node degrees and triangle participation.

Finally, I remind the authors that triangle participation is by definition a topological - not geometrical - property, and in principle, there can simultaneously exist several sources for the appearance of triangles in networks. The authors nicely illustrate one source: a latent geometry. However, it is possible for other sources to exist, such as dynamical processes on the network (e.g., processes for triangle closure that are independent of the metric space). Therefore, the correlation between triangles and weights (e.g., Fig. 1) can indicate a relation between a latent geometry and weights, or it can simply indicate a correlation between triangles and weights (that is, one could argue that the latent-geometry origin of triangles is superfluous).

This issue should be addressed in the paper, and I leave it to the authors to decide 'how' to do this. I can suggest some possible extensions that may help support claim (ii). First, I suspect Fig. 3d can be interpreted as a measure for determining whether the correlation between triangles and weights is a fundamental relationship, or if it is an artifact of a latent geometry. Specifically, for E. Coli and the brain, it appears that the correlation between triangles and edge weights can be explained entirely by the latent geometry. If the authors agree, then this should be discussed. Second, I would urge the authors to conduct a small simulation to compare their latent-geometry model to a simpler model in which edge weights only depend on node degree and triangle participation. I believe it would be interesting (and very strong evidence) if TIV curves can discriminate whether the organization of edge weights is better explained by triangle participation or by a latent geometry.

(3.C) The authors do a good job of further describing potential applications of their work. They may find it helpful to briefly discuss the implications of this work toward previous research on link

prediction, since triangle participation is widely-adopted as a leading approach:

Liben-Nowell, D., & Kleinberg, J. (2007). The link-prediction problem for social networks. *Journal of the American society for information science and technology*, 58(7), 1019-1031.

Lü, L., & Zhou, T. (2010). Link prediction in weighted networks: The role of weak ties. *EPL (Europhysics Letters)*, 89(1), 18001.

Zhao, Jing, Lili Miao, Jian Yang, Haiyang Fang, Qian-Ming Zhang, Min Nie, Petter Holme, and Tao Zhou. "Prediction of links and weights in networks by reliable routes." *Scientific reports* 5 (2015).

Importantly, the last method specifically aims to predict edge weights, and so the authors' claim that their method 'for the first time, will provide estimates for the weights of such missing links' is false. In fact, the authors do not actually use their method to do link prediction, so this claim about a potential application is an overstatement.

(3.D). The authors have chosen to still not provide a technical description in the paper for how clustering arises for their new model. Sec. II begins: 'Clustering, as a reflection of the triangle inequality, is the key topological property coupling the bare topology of a complex system and its effective underlying metric space [6]. In this context, the triangle inequality stipulates that if nodes A and B are close, and nodes A and C are also close, we expect nodes B and C to be close as well; triangles are therefore more likely to exist between nodes that are nearby.'

This extremely simplistic explanation is appropriate in Sec. II since the authors have not yet defined their model. However, a similarly simplistic explanation is again stated in Sec. IV.A (even after the model is introduced). At this point, I would have found a more technical description for the appearance of triangles very helpful. If the derivation is identical to that in [6], it would be helpful to point to the relevant equations in [6] (of course, this requires the notation to be identical), otherwise I suggest including a brief summary in Sec. IV or an appendix. For example, I believe it would be helpful to include some of the discussion in the authors' second paragraph of their response to my comment (3.E).

(3.E). I appreciate the more-in-depth description, which will allow me to more precisely state my main concern, which was not addressed in the authors' response.

Specifically, if the latent-geometry model implies that the weight w_{ij} of edge (i,j) depends on the variable $\psi_{ij} = \Delta_{ij}/\kappa_j \kappa_j$, then the accuracy and inference of the model can be directly explored by studying the relationship between these two variables. Instead, the authors study the nature of edge weights w_{ij} through studying triangles. Again, with Occam's razor principle in mind, it is important that the authors provide evidence and explain in the manuscript why it is beneficial to validate and fit their model using the more complicated approach of studying triangles versus the simpler approach of studying edges.

In other words, the most direct way to determine if there is a relationship between the latent geometry distances x_{ij} and weights w_{ij} is simply to compare these - why resort to studying triangle inequalities?

As a related comment: In their response to my previous comment (3.E), the authors write "Rather, it is an abstract distance that quantifies the likelihood of interactions between nodes. Consequently, a direct measurement is not available ... " I am confused why a direct measurement is not available. If one constructs an embedding, then one has x_{ij} .

Issues regarding new material:

Paragraph just before II: "This model has the critical ability to discriminate between purely local properties (e.g., related to the degree and strength of nodes) and the coupling of the topology and of the weighted organisation with the metric space." -- I would say that triangle participation is a local property too; it depends only on a node and its neighbors. Is local vs. nonlocal really the focus of the paper or is it geometric vs non-geometric?

1. Replies to the comments of Reviewer #1

1.A *After having read the new version of the manuscript and the authors' responses to the reviewers' remarks, I remain skeptical about the significance of these results and their suitability for Nature Communications. As a consequence, I still do not recommend publication of this manuscript.*

We thank the reviewer for his/her comments on our manuscript. Below, we provide detailed responses to all of his/her criticisms and hope that the reviewer will be convinced by our arguments.

1.B *In their reply 1.E, the authors write "...the concept of a hidden metric space can also be seen as a mathematical tool that can be leveraged to generate realistic networks. For instance, it is the only framework that allows to generate strong clustering based on pairwise interactions only (due to the triangle inequality of the metric space)".*

This is not true: even random graphs with given degrees can have a large clustering. This is not often recognised, but dates back to Park and Newman's PRE paper "Origin of degree correlations in Internet and other networks". The reason why this result is overlooked is the widespread use of the approximation that factorizes the connection probability into the product of the end-point degrees. As originally showed by Maslov, Sneppen and collaborators, this approximation is inconsistent with the large value of the maximum degree in real-world networks. If realistic degree sequences are to be replicated, one needs to go beyond the naive factorized approximation. The resulting probability of connection is highly nonlinear (it has a Fermi-function shape) and was derived by Park and Newman in the paper above and in many subsequent papers. This probability function is the correct one for a network with broad degree distribution and generates a high level of clustering (often matching perfectly the empirical clustering), even if it only accounts for local (degree) properties of nodes, without resorting to any metric pairwise property.

Therefore the apparent need to introduce metric spaces to replicate high clustering might be merely an artefact of the (incorrect) approximation of the connection probability. Note that, even if it is often said that the factorized probability works well for "sparse networks", this is actually incorrect: a factorized probability does generate sparse networks, but these networks are however unrealistic in terms of their maximum degree. In other words, real-world networks, although sparse in most cases, are incompatible with the factorized approximation. Compensating this unrealistic approximation with the introduction of a metric space in order to retrieve an otherwise unexplained large clustering is scientifically incorrect and misleading. The large clustering (or at least a generous portion of it) would more parsimoniously be explained by using only local properties (e.g. degrees), along with the correct nonlinear connection probability accounting for them.

We partly agree with the reviewer in that heterogeneous degree distributions generate clustering. However, his/her complain in this regard is a bit paradoxical given that one of us wrote a paper in 2013 precisely calculating the clustering coefficient of scale-free networks under the configuration model, by explicitly considering the connection probability derived in the work by Park and Newman [see Phys. Rev. E 86, 026120 (2012)]. In that work, we showed that, indeed, clustering can be important in heterogeneous random graphs with γ close to 2. However, we also showed that clustering always vanishes in the thermodynamic limit (even though slowly in some cases). In any case, the reviewer is confused about the origin of clustering as a result of the non-factorization of the connection probability. In fact, it is rather the opposite as the formula for the clustering coefficient obtained by using the factorized connection probability [see Eq. (1) in Phys. Rev. E 86, 026120 (2012)] leads to an overestimation of the clustering coefficient in general and to a diverging clustering coefficient for $\gamma < 7/3$, a result that is obviously wrong. The non-factorized connection probability arises as a consequence of the closure of the network when there are degrees above \sqrt{N} in the network, leading to (negative) structural correlations and, incidentally, to the correct expression for the clustering coefficient, which is obviously non-diverging.

In any case, while some portion of the clustering observed in real networks could be explained by these finite size effects, it is typically much higher than the clustering observed in randomized versions of the same networks. To give support to this statement, we have randomized many real world networks, including those used in our study, by preserving, in one case, the degree distribution and, in a second case, the degree distribution and also the degree-degree correlations of the real networks. Randomizations are performed using the software developed in Scientific Reports 3, 2517 (2013) and Nature Communications 6, 8627

Figure 1: Average local clustering coefficient measured for original real networks and their randomized counterparts. Both randomizations, CM and CCM, preserve the degree distribution and the CCM additionally preserves the degree-degree correlations. Results for the randomized networks were obtained by averaging over 100 realizations and the error bars show the 5th and 95th percentiles.

(2015). Figures 1 and 2 show the results. We observe that, in all networks, the clustering coefficient is much larger than in the randomized versions (by more than three sigmas). These results are also valid in most of the real networks we are aware of.

In the light of these results, it is thus important to have models able to explain clustering that remain high and finite in the infinite size limit. In this respect, metric spaces (hidden or not) underlying complex networks provide the simplest explanation for its origin. The reason is that metric spaces induce many body interactions out of pairwise interactions only. In a network, one can think of triangles as some evidence of three body interactions among the elements of the network. We are then faced with only two possibilities, either we have a mechanism with genuine three (or more) body interactions, which is a priori unknown, or we assume the existence of a metric space. In our opinion, the latter option is the simplest and most natural. It also allows for analytic tractability and, thus, the ability to compare with real systems. In this respect, we would like to mention our result in Phys. Rev. Lett. 100, 78701 (2008), where we show that the self-similarity properties of several real complex networks can be accounted for with the hypothesis of hidden metric spaces underlying the networks. Besides, when mapping real networks into our models, like the Internet [Nature Communications 1, 62 (2010)], metabolic networks [Molecular BioSystems 8, 843-850 (2012)], or the world trade web [Scientific Reports 6, 33441 (2016)], and compare their embeddings with metadata not included in the graph itself, like country affiliation or biological pathway, we find a very strong congruency, suggesting that our embeddings are not an artifact of the method and reflect the real organization of these systems.

As a final note, during these years working in the field of complex systems and complex networks, we have gained a solid reputation as serious scientists. In particular, the quality of our studies about the connection between the topology of complex networks and hidden metric spaces is beyond doubt in the community and our works have been published in leading international journals including Nature, Nature Physics, Nature Communications, Physical Review Letters, and others. Therefore, we would like to ask the reviewer to refrain from using statements of the type "...scientifically incorrect and misleading." about our work.

1.C Even though such a factorized approximation is never introduced explicitly as a building block of the model described in this paper, an equivalent problem is present here in terms of the expected weight being linearly dependent on the expected strengths. Indeed, as a consequence of this approximation, in the model proposed the expected strengths turn out to be proportional to the corresponding hidden variables, apparently justifying the claim that their model can account for any (joint) degree and strength distribution. Again, this claim of generality is not founded and the resulting metric "patterns" might be

Figure 2: Degree-dependent average local clustering for various real networks. See the caption of Fig. 1 for a description of the randomization procedures.

an artefact compensating for the factorised choice of the expected weights.

We should stress that, in our model, weights do not factorize because the distance between two nodes in the metric space cannot be factorized. The reviewer may have in mind another of our previous works [Phys. Rev. E 74, 055101(R) (2006)], where we show that weighted networks have structural correlations. It is important, however, to realize that such structural correlations appear when one considers actual degrees and strengths of nodes, and not their expected values. This is quite different from the case of the bare topology. To generate connections in a graph, the connection probability must be bounded between zero and one, and thus the connection probability cannot be factorized even at the level of hidden variables (or expected values) in strongly heterogeneous networks. In the case of weights, there is no such restriction and expected weights among nodes can be defined in an arbitrary way. Nevertheless, structural correlations at the weighted level will appear due to structural constraints (see Fig. 1 in PRE 74, 055101(R) (2006)).

As for our claim about the ability of our model to generate networks with desired correlations between strength and degree, we first notice that, in Eq. (11) of the Supplementary Information, we provide the exact probability for a node with hidden variables κ and σ to have degree and strength k and s , respectively.

Combining this result with the joint distribution of hidden variables $\rho(\kappa, \sigma)$, we obtain the joint degree-strength distribution. Given that we have complete freedom to choose $\rho(\kappa, \sigma)$, we can control the level of correlations between k and s , as claimed in the paper. In particular, we can choose $\sigma \propto \kappa^\eta$, which translates into $\bar{s}(k) \propto k^\eta$, as corroborated by our numerical simulations shown in the Supplementary Information. Note that our model can actually generate networks with any value of the exponent η , even if $\eta < 1$ (see Fig. S13), something that, to the best of our knowledge, cannot be accomplished with other models of weighted networks.

1.D Additionally, since the authors want to decouple local node effects (degrees and strengths) from the (postulated) metric properties, it is not clear why they preliminarily filter most of their networks with the “disparity filter” (by the way, why don’t they do this for all networks? In the SI they say that some networks have been filtered this way, and others not, without explanation). By using this filter, the local effects should in principle vanish, so they should be left with “residual networks” where local node properties are no longer relevant and should not be further controlled for. So why are they applying their model to these filtered networks? This procedure is unclear to me. In any case, it raises the doubt whether the empirical “patterns” that are documented here are actually properties of how the disparity filter operates, rather than properties of the data themselves.

The reason to use the disparity filter in some of the networks is related to the huge average degree of some of these networks and the fact that most of the links contributing to such large average degree are not significantly related to the main functionality of the network. For instance, in the US airport network, there are a huge number of connections between airports with a number of seats of the order of tens during one whole year. All these connections are there due to private flights that, obviously do not follow the same patterns of connections of commercial (and so regular) airline connections between airports. The same applies to other networks, like the world trade web, where we find an enormous number of trade interactions between countries with a total amount traded of the order of 1 million dollars or less. Such trade interactions are extremely volatile and appear and disappear every year and cannot represent a solid trade interaction between the countries to be exploited in the recognition of characteristic interaction patterns. The disparity filter is extremely good at removing such noisy links, revealing the fundamental structure of the system. It is not true that, by using the filter, local effects vanish or that it leads to residual networks. In fact, in PNAS 106, 6483-6488 (2009), we showed that the disparity filter retains a significant fraction of the total weight in the system without altering the local properties (e.g. clustering), the non-linear correlations between strength and degree, and the strength and weight distributions, while reducing significantly the number of links and, thus, revealing the fundamental degree distribution of the network. In this respect, it is interesting to mention one of the results in our last paper on the world trade web [Scientific Reports 6, 33441 (2016)]. In this work, we measured the correlation between degrees and countries’ gross domestic product (GDP) for the original network and for the network filtered by our disparity filter. Interestingly, the Pearson correlation coefficient in the case of the original network is of the order $0.2 \sim 0.4$ depending on the year, whereas for the filtered network takes values of the order $0.8 \sim 0.9$. This simple test indicates that the topology of the filtered network is significantly more congruent with real economic factors than the original one. In the new version of the manuscript, we have added a similar discussion about the use of the disparity filter.

1.E By the way, the disparity filter assumes that the total strength s of a node is uniformly randomly broken up into the weights of the k edges coming out of a node, irrespective of the degrees at the other endpoint of these edges. This again appears to contrast the well-known fact, used also elsewhere in this paper, that connection probabilities should depend on the degrees at both endpoints of an edge. So here I see some inconsistency in the way data are analysed.

Please notice that, in the disparity filter, the relevance of a link is addressed from both end nodes, and that it is removed only if it is deemed irrelevant for both of them. This procedure restores the symmetry the reviewer was concerned about.

1.F Finally, it is not true that this is the first “generation of gravity models” assuming that also the probability of connections should be a gravity-like function. There is vast literature about the so-called zero-inflated gravity models which do have a similar dependence of link probabilities on the gravity equation, thus replicating the observed network density (see for instance the published papers by Fagiolo (<http://arxiv.org/abs/0908.2086>) and Fagiolo and Duenas (<https://arxiv.org/abs/1112.2867>) and refer-

ences therein.

We thank the reviewer for pointing out these interesting contributions. We would like to stress that nowhere in the manuscript do we claim that it is the “first generation of gravity models”; we rather state that our model offers a “new generation of gravity models”. The only “first” claimed in the manuscript concerns the possibility to provide estimates of the weights of missing links in the framework of embeddings of real networks. This wording was a bit misleading and has been modified accordingly. In the new version of the manuscript, we have added a citation to the work by Fagiolo and Dueñas.

In any case, previous works were not successful in replicating simultaneously the weighted structure and the topology of complex networks using gravity models, as explicitly recognized for instance in one of the papers pointed out by the referee, Fagiolo and Dueñas (<https://arxiv.org/abs/1112.2867>): “*More generally, the GM performs very badly when asked to predict the presence of a link, or the level of the trade flow it carries, whenever the binary structure must be simultaneously estimated. Therefore, the GM turns out to be a good model for estimating trade flows, but not to explain why a link in the ITN gets formed and persists over time*”. In contrast, our framework, based on gravity models both for the weights and for the existence of links, not only reproduces well the weighted structure of complex networks but also their topological properties, much beyond the network density which is a very rough topological feature easily reproducible if other topological properties are overlooked.

1.G In conclusion, I still believe that this papers does not introduce a really new and general paradigm to explain the origin of weights in real networks. It might be forcing the use of metric spaces to compensate for some implicit proportionality assumption (for sure it is partially doing so), it might be partially looking at spurious patterns created by the filtering method used, and it is not the first/only model that has been proposed to understand the empirical weights in weighted networks.

We hope we have clarified all reviewer's concerns.

3. Replies to the comments of Reviewer #3

I have examined the authors' revised manuscript and their responses to my comments. Although several of my concerns have been adequately addressed, the authors did not directly address several of the main issues that I previously raised.

As I previously stated, my overall feeling is that the paper provides a nice contribution to this field and deserves publication in some form and at some venue. However, I cannot support publication in Nature Communications until the issues below are carefully addressed in the manuscript. That said, I now believe the manuscript to be sufficiently impactful to warrant publication in Nature Communications. My recommendation is now 'revise and resubmit.'

We thank the reviewer for his/her positive opinion about our work and for the very helpful and constructive comments to improve the quality and presentation of our paper. In the new version of the manuscript, we have followed his/her recommendations and we hope that all the reviewer's concerns are fully addressed and clarified.

3.A I believe the authors missed my main concern, which regards my previous statement “It is unclear whether or not the hyperbolic geometry modeling approach provides further insight than what is possible by studying the dependence of weights on node degrees and triangle participation (i.e., neighbourhood overlap).” I will further explain this concern.

Specifically, given the observations that node degree and triangle participation both influence edge weights, the simplest model would be one in which weights depend only on two types of variables: node degrees and triangle participation. My concern regards whether or not the complicated latent-geometry model satisfies the Occam's razor principle. I believe that it does, but given the complexity of their model, the authors should provide strong evidence and a clear discussion for why the hyperbolic-geometry model is superior to a simpler alternative.

I point out that a correlation between triangle participation and edge weight is widely believed, despite –

as identified by the authors – some results in [19] are lacking evidence since they do not isolate the effect some of their experiments. I agree that the authors conduct a more principled experiment with Fig. 1, but the main message of Fig. 1 (i.e., triangles influence edge weights) is not a new idea. It is actually the focus of [19], which is a paper that includes more results than the single experiment upon which the authors improve.

The authors' novel claim with Fig. 1 is that a hidden geometry is the origin of this phenomenon. That is, the correlation between triangle participation and edge weight is (or can be) an artifact of a correlation between geometry and edge weight. I believe such a claim requires two types of support:

(i) The hidden geometry model can account for the correlation between edge weight and triangle participation.

(ii) The hidden geometry model provides a “better” explanation versus a much simpler model in which edge weights only depend on node degrees and triangle participation.

(i) is strongly supported by their study of TIV curves. In my opinion, (ii) is insufficiently described in the paper. That is, the authors do not clearly explain why adopting a complicated latent-space model for edge weights is superior to a simpler alternative model in which one only takes into account node degrees and triangle participation.

Finally, I remind the authors that triangle participation is by definition a topological - not geometrical - property, and in principle, there can simultaneously exist several sources for the appearance of triangles in networks. The authors nicely illustrate one source: a latent geometry. However, it is possible for other sources to exist, such as dynamical processes on the network (e.g., processes for triangle closure that are independent of the metric space). Therefore, the correlation between triangles and weights (e.g., Fig. 1) can indicate a relation between a latent geometry and weights, or it can simply indicate a correlation between triangles and weights (that is, one could argue that the latent-geometry origin of triangles is superfluous).

This issue should be addressed in the paper, and I leave it to the authors to decide “how” to do this. I can suggest some possible extensions that may help support claim (ii). First, I suspect Fig. 3d can be interpreted as a measure for determining whether the correlation between triangles and weights is a fundamental relationship, or if it is an artifact of a latent geometry. Specifically, for *E. Coli* and the brain, it appears that the correlation between triangles and edge weights can be explained entirely by the latent geometry. If the authors agree, then this should be discussed. Second, I would urge the authors to conduct a small simulation to compare their latent-geometry model to a simpler model in which edge weights only depend on node degree and triangle participation. I believe it would be interesting (and very strong evidence) if TIV curves can discriminate whether the organization of edge weights is better explained by triangle participation or by a latent geometry.

Thank you very much for these insightful thoughts, which are indeed very pertinent. First, we would like to comment on our own point of view about the Occam's razor principle. We fully agree with this principle: simpler models should be preferred over more complicated ones if they are able to explain the same empirical facts. However, in this case, it is not totally clear the meaning of a model being simpler than another one. In the case of clustering, for instance, triangles in networks can be interpreted as the signature of three body interactions. To explain this empirical fact, we have two options, either we model the system with a genuine mechanism inducing three body interactions, which is unknown, or alternatively we assume the existence of an underlying metric space combined with pairwise interactions, which induces many body interactions. The question is now: is a model with pairwise interactions on a metric space more complicated than a model with many body interactions without a metric space? In our opinion, metric spaces combined with pairwise interactions are a much simpler explanation for the topologies of real complex networks. Also from the mathematical point of view, this possibility is more interesting as pairwise interactions allow for analytical treatment and, thus, a much simpler comparison with empirical data with a limited number of model parameters. Of course, this discussion mainly applies to the bare network topology and not to the weights. However, if we accept the existence of such metric space as an explanation for the network topology, it seems also reasonable that the same metric space will, somehow, influence the intensity of the interactions.

As for the specific suggestions of the reviewer about a model that would use only topological information,

we agree that this is certainly a good exercise. However, it is a difficult task, since the number of such models one could define is very large. Paradoxically, the literature is not very generous in terms of credible models suitable for such exercise (e.g., that do not rely on intricate dynamics to assign the weights and, as such, would not pass the Occam's razor principle). In any case, we have opted for the models used in [Nature Physics 8, 429 (2012) and in PNAS 101, 3747 (2004)] and for a new one that generalizes both. If we understand correctly the reviewer's suggestion, the idea is to explain the observed weighted network structure without relying on any metric space whatsoever. Therefore, we first randomize the real network topologies preserving the degree sequence and the average clustering coefficient [for this task, we use the software developed in Scientific Reports 3, 2517 (2013) and Nature Communications 6, 8627 (2015)]. This step is taken to destroy any dependence on any possible metric space underlying the network. Then, we assign weights to the connections according to the following models:

- **model A:** $w_{ij} \propto (k_i k_j)^\theta$, where k_i and k_j are the degrees of nodes i and j , respectively, and θ is a model parameter;
- **model B:** $w_{ij} \propto (c_i c_j)^\delta$, where c_i and c_j are the clustering coefficient of nodes i and j , respectively, and δ is a model parameter;
- **model C:** $w_{ij} \propto (k_i k_j)^\mu (c_i c_j)^\nu$, and μ and ν are model parameters. This model accounts for the fact that weights among high degree nodes are higher but also that weights among highly clustered nodes are also higher.

For all models, the exponents θ , δ , μ and ν are chosen as those minimizing the χ^2 statistic for the corresponding dataset, and the results are shown in Figs. S14-S41 in the Supplementary Material. Although the three models preserve the degree sequence, which is an advantage over our model, the degree-dependent clustering of the synthetic networks is worse reproduced as compared to the one obtained with our model. We find that models A and C can reproduce fairly well the strength distribution, or at least its general shape. This is due to the strong influence of the topology over the weighted organization, and it illustrates well the reason why we factorized the weights in Fig. 1 to account for the effect of the topology. However, except for the world trade web and the US airports network, we find that the three models reproduce poorly the distribution of weights and the disparity. This is not particularly surprising in the case of the US airports network since this is the network for which our model predicts a weaker dependence on the metric space, leaving weights mainly a function of nodes' degree. It is not surprising in the case of the world trade web either, given the small size of the network and the strong fluctuations present on it. Nevertheless, our model is the only one that consistently reproduces the properties of the real weighted networks with accuracy. More importantly, the three models perform very badly at reproducing the triangle inequality curves for all networks (as shown on Figs. S14-S41). As pointed out by the referee, this provides a very strong evidence that our assumption about the metric origin of weights is a much better explanation of the real data. We have added a new section in the Supplementary Information to include this new analysis.

3.B The authors do a good job of further describing potential applications of their work. They may find it helpful to briefly discuss the implications of this work toward previous research on link prediction, since triangle participation is widely-adopted as a leading approach:

Liben-Nowell, D., & Kleinberg, J. (2007). The link-prediction problem for social networks. Journal of the American society for information science and technology, 58(7), 1019-1031.

Lü, L., & Zhou, T. (2010). Link prediction in weighted networks: The role of weak ties. EPL (Europhysics Letters), 89(1), 18001.

Zhao, Jing, Lili Miao, Jian Yang, Haiyang Fang, Qian-Ming Zhang, Min Nie, Petter Holme, and Tao Zhou. "Prediction of links and weights in networks by reliable routes." Scientific reports 5 (2015). Importantly, the last method specifically aims to predict edge weights, and so the authors' claim that their method "for the first time, will provide estimates for the weights of such missing links" is false. In fact, the authors do not actually use their methods to do link prediction, so this claim about a potential application is an overstatement.

We thank the referee for pointing out interesting publications (especially the third one which had somehow slipped under our radar) and we agree that our choice of wording is a bit ambiguous. Embeddings of

unweighted networks in metric spaces have already been shown to permit the prediction of missing links [see for instance Nature 489, 537-540 (2012)], and we simply meant that our model now allows to extend this powerful methodology to weighted networks. In other words, “for the first time” was referring to “the first time” in the framework of networks embedded in metric space. We have corrected this ambiguity in the main text and added these new references.

3.C The authors have chosen to still not provide a technical description in the paper for how clustering arises for their new model. Sec. II begins: “Clustering, as a reflection of the triangle inequality, is the key topological property coupling the bare topology of a complex system and its effective underlying metric space [6]. In this context, the triangle inequality stipulates that if nodes A and B are close, and nodes A and C are also close, we expect nodes B and C to be close as well; triangles are therefore more likely to exist between nodes that are nearby.”

This extremely simplistic explanation is appropriate in Sec. II since the authors have not yet defined their model. However, a similarly simplistic explanation is again stated in Sec. IV.A (even after the model is introduced). At this point, I would have found a more technical description for the appearance of triangles very helpful. If the derivation is identical to that in [6], it would be helpful to point to the relevant equations in [6] (of course, this requires the notation to be identical), otherwise I suggest including a brief summary in Sec. IV or an appendix. For example, I believe it would be helpful to include some of the discussion in the authors’ second paragraph of their response to my comment (3.E).

Thank you very much for this suggestion. In the previous version of the manuscript, we decided not to include such details because they have already been discussed in our previous publications. However, we agree that by adding such discussion the paper becomes more self-contained. Therefore, in the new version of the manuscript we have included the discussion mentioned by the reviewer in Sec. IV.

3.D I appreciate the more-in-depth description, which will allow me to more precisely state my main concern, which was not addressed in the authors’ response.

Specifically, if the latent-geometry model implies that the weight w_{ij} of edge (i, j) depends on the variable $\psi_{ij} = \Delta_{ij}/\kappa_j\kappa_j$, then the accuracy and inference of the model can be directly explored by studying the relationship between these two variables. Instead, the authors study the nature of edge weights w_{ij} through studying triangles. Again, with Occam’s razor principle in mind, it is important that the authors provide evidence and explain in the manuscript why it is beneficial to validate and fit their model using the more complicated approach of studying triangles versus the simpler approach of studying edges.

In other words, the most direct way to determine if there is a relationship between the latent geometry distances x_{ij} and weights w_{ij} is simply to compare these - why resort to studying triangle inequalities?

As a related comment: In their response to my previous comment (3.E), the authors write “Rather, it is an abstract distance that quantifies the likelihood of interactions between nodes. Consequently, a direct measurement is not available ...” I am confused why a direct measurement is not available. If one constructs an embedding, then one has x_{ij} .

There are some subtle issues here. We can, of course, find an embedding of a given network and then compare weights directly against x_{ij} . This is, indeed, the main idea behind the plot in Fig. 3d. However, to do so, we use a statistical inference technique that relies on the assumption that the network topology has been generated by the model. Therefore, one could argue that, since we are fitting the data to the model, it is not surprising to find a metric relation with the weights. This is, of course a misleading concern because to find the embeddings we do not use information from the weights. In any case, for the moment, we only know how to do embeddings of unweighted networks. In this manuscript, we are precisely proposing the geometric model for weighted networks, which is the first step needed to propose an embedding method for weighted networks in the future. The consideration of weights in the embedding process will certainly change the inferred coordinates of the nodes, so that distances inferred from the topology, although significantly correlated with weights, are not enough to explain the weighted structure of networks, as expected. For these two reasons, we wanted a method that would be able to find metric dependencies without relying on any embedding, that is, using only the network topology and the actual weights. We find that our method to compute the triangle inequality curves fits this purpose very well. In

the new version of the manuscript, we have tried to be clearer in this respect.

3.E Paragraph just before II: "This model has the critical ability to discriminate between purely local properties (e.g., related to the degree and strength of nodes) and the coupling of the topology and of the weighted organisation with the metric space." – I would say that triangle participation is a local property too; it depends only on a node and its neighbors. Is local vs. nonlocal really the focus of the paper or is it geometric vs non-geometric?

Perhaps we have not been very clear with this sentence. What we mean is that our model allows us to fix the degree-strength distribution independently of the coupling with the metric space. Therefore, it allows us to discount the effects of the degree-strength structure to reveal the coupling with the metric space in real networks. Imagine, for instance, a model without this property, one where by changing the coupling with the geometry we would obtain a completely different degree-strength distribution. Such model would be extremely difficult to contrast against real weighted networks. In the new version of the paper, we have rephrased this paragraph to clarify its meaning.

REVIEWERS' COMMENTS:

Reviewer #1 (Remarks to the Author):

I am very sorry if the authors got offended by my use of words such as "misleading" or "incorrect". My intention was not that of offending, of course, but that of pointing out various aspects of this research that may indeed lead to confusing interpretations, and that in my view do not justify its publication in a non-specialised and broad-audience journal like Nature Communications.

I also did not want to cast doubts on how respected some of the authors in this manuscript are internationally (I know they are). I just wanted to point out that these authors have carried out much better research in the past, and that this particular work does not raise particular scientific interest in my opinion.

Since I have been asked to have a final critical look at the revised manuscript, I hope I can better explain the main reasons for my judgement in this report. I am not going to go over all the points of my criticism again (from the authors' last reply, it is clear that we disagree at many points); I just want to emphasise the main concerns, which do survive and remain serious after the authors' revisions to the manuscript.

First of all, I hope the authors will agree that the value of their work has to be assessed NOT in relation to the mathematical model itself (proposing an abstract mathematical model of embedded weighted networks is certainly interesting but not exciting, and does not per se deserve publication in Nature Communication), BUT to the degree to which such model can explain the empirical structure of real-world networks. So I should not judge their results for synthetic networks, as these results are irrelevant for the value of the paper. This restricts the relevant assessment of the paper to the last four figures, i.e. the single figure with the triangle inequality violation (TIV) curve for the 3 real networks and the three 6-panel figures with the distributions/spectra of various topological networks properties.

When it comes to this crucial empirical analysis, my concerns are really serious. The consistency between the model and real networks is not convincingly studied, in my opinion, for the following reasons.

1) Only three networks are analysed, while the authors claim that their mechanism might be general and explain the nature of weights in generic real-world networks. Replicating the analysis AS IT IS on more networks would however still not be enough, because the tests of the consistency between model and data are unsatisfactory, due to the other two reasons below.

2) In the 6-panel figures, the consistency is studied only in terms of overall properties like the degree distribution, the strength distribution, the strength-degree relationship, the weight distribution, the disparity, and the clustering-degree relationship. Now, for the first three properties the agreement is totally unsurprising, given that their model (as they repeatedly mention throughout the paper) can control for any degree distribution, any strength distribution, and any form of degree-strength correlation.

Coming to the remaining three properties (weight distribution, disparity and clustering), my main concern is that, for many real networks, it turned out that these properties (or very similar ones) can be explained very well even WITHOUT invoking any coupling to an underlying metric space. See for instance the works by Garlaschelli and coauthors about maximum-entropy models of weighted networks, where it was shown that many properties of real weighted networks (including the WTW studied here) can be explained on the basis of strength and degrees alone. (By the way, I now realise that a recent extension of these models to the case of distance-dependent networks, <http://arxiv.org/abs/1506.00348>, appeared prior to the manuscript under review here but is not cited).

Now, given that the model proposed here has an extra parameter (the coupling α with the postulated metric space), and that this parameter has a special value for which no coupling is

realised, it is obvious that, purely because of the presence of an extra parameter, the model can fit the data better than a model without such a parameter, but where the degrees and strengths can be equally controlled for. Moreover, the agreement with the empirical weight distribution is not a strong test of their model, as one would like the latter to replicate (modulo the noise) each individual weight one by one, and not the statistical distribution alone (the weight distributions of the model and the data can be identical even if no single weight is correctly replicated by the model).

This leads me to the conclusion that the 6-panel figures are not conclusive about the agreement between the model and the data: simpler models without the metric hypothesis may lead to equally good results.

3) The only remaining real test of their hypothesis is the TIV curve. Now, I did not realise in my first reading of the manuscript that this quantity only tests the triangular inequality on the REALISED TRIANGLES in the network. It is clear that this restricts the analysis to the triples of nodes for which, at a purely topological level, the triangular inequality is already most likely to be realised. So the TIV (which is the ratio of the REALISED triangles that violate the triangular inequality to the total number of REALISED triangles) is a very weak measure of their hypothesis. I understand that the authors propose a sort of separation between the topology (which is pre-determined assuming a metric coupling) and the weights (which are established on the realised links, again assuming a metric coupling). It is however difficult to become convinced that one should not base the analysis of the violation of triangular inequality in real weighted networks to ALL the triples of nodes, including those that are not realised triangles.

Clearly, if V-shaped triples of nodes are included in the analysis, the violation of triangular inequality can presumably only get much bigger, leaving us little room to believe that hidden metric spaces are indeed at play behind real weighted networks. Note that, while an analysis of V-shapes (or wedges) would be not so informative for binary networks, it would be very informative for weighted networks, given that I expect many V-shapes with two links with a strong weights (e.g. two peripheral nodes connected to the same hub) and a missing third link (between the two peripheral nodes). These weighted patterns appear to be in stark contrast with the metric hypothesis for weighted networks, and the fact that they are omitted in this analysis can be quite deceptive (again, no offence meant).

Reviewer #3 (Remarks to the Author):

I have reviewed the manuscript "The geometric nature of weights in real complex networks" for the third time, and now recommend publication of the manuscript in Nature Communications. The authors have carefully and adequately addressed the concerns I previously raised.

Overall, I find the paper to be a pioneering contribution for hyperbolic embeddings of weighted networks, a very important topic for network science.

1. Replies to the comments of Reviewer #1

1.A *I am very sorry if the authors got offended by my use of words such as "misleading" or "incorrect". My intention was not that of offending, of course, but that of pointing out various aspects of this research that may indeed lead to confusing interpretations, and that in my view do not justify its publication in a non-specialised and broad-audience journal like Nature Communications. I also did not want to cast doubts on how respected some of the authors in this manuscript are internationally (I know they are). I just wanted to point out that these authors have carried out much better research in the past, and that this particular work does not raise particular scientific interest in my opinion.*

Since I have been asked to have a final critical look at the revised manuscript, I hope I can better explain the main reasons for my judgement in this report. I am not going to go over all the points of my criticism again (from the authors' last reply, it is clear that we disagree at many points); I just want to emphasise the main concerns, which do survive and remain serious after the authors' revisions to the manuscript.

We thank the referee for his/her last report although we are sorry for not being able to convince him/her about the importance of our work. Below, we provide detailed responses to the last comments by the referee that we hope will help to clarify all his/her doubts about our work.

1.A *First of all, I hope the authors will agree that the value of their work has to be assessed NOT in relation to the mathematical model itself (proposing an abstract mathematical model of embedded weighted networks is certainly interesting but not exciting, and does not per se deserve publication in Nature Communication), BUT to the degree to which such model can explain the empirical structure of real-world networks. So I should not judge their results for synthetic networks, as these results are irrelevant for the value of the paper. This restricts the relevant assessment of the paper to the last four figures, i.e. the single figure with the triangle inequality violation (TIV) curve for the 3 real networks and the three 6-panel figures with the distributions/spectra of various topological networks properties.*

We fully agree with the referee that the value of our work is not in our model but on the fact that it can explain very well the patterns that we observe in real networks. We are, however, a bit surprised about the description that the referee makes of our work. Our manuscript starts with an empirical analysis of seven (not three) real weighted networks from very different domains. This is shown in figure one, where we show that there is a different weighted organization of links that participate in triangles with respect to those that do not participate in triangles. We interpret this empirical finding as a signature of an underlying metric space and, then, we introduce our geometric model to explain such empirical observations. Our work is not focused on the model as the referee suggests, even though we strongly believe that our model is, at present, the best model for weighted networks in the market. We do not understand either why the referee talks about the last four figures because our manuscript has only four figures. Besides, the single figure with the TIV curve is actually a 4-panel figure with seven (not three) real networks and we only show one 6-panel figure with the various topological networks properties, the rest of 6-panel figures are included in the Supplementary Information.

1.A *When it comes to this crucial empirical analysis, my concerns are really serious. The consistency between the model and real networks is not convincingly studied, in my opinion, for the following reasons.*

1) *Only three networks are analysed, while the authors claim that their mechanism might be general and explain the nature of weights in generic real-world networks. Replicating the analysis AS IT IS on more networks would however still not be enough, because the tests of the consistency between model and data are unsatisfactory, due to the other two reasons below.*

Please notice that we analyse seven different real weighted networks from very different domains and not three.

1.A 2) *In the 6-panel figures, the consistency is studied only in terms of overall properties like the degree distribution, the strength distribution, the strength-degree relationship, the weight distribution, the disparity, and the clustering-degree relationship. Now, for the first three properties the agreement is totally unsurprising, given that their model (as they repeatedly mention throughout the paper) can control for any degree distribution, any strength distribution, and any form of degree-strength correlation.*

Of course, these measures were included as a consistency check of the model so that we are sure that, indeed, our model does what is claimed in the manuscript it does, that is, to have full control of the joint degree-strength distribution, regardless of the level of coupling with the metric space.

1.A Coming to the remaining three properties (weight distribution, disparity and clustering), my main concern is that, for many real networks, it turned out that these properties (or very similar ones) can be explained very well even WITHOUT invoking any coupling to an underlying metric space. See for instance the works by Garlaschelli and coauthors about maximum-entropy models of weighted networks, where it was shown that many properties of real weighted networks (including the WTW studied here) can be explained on the basis of strength and degrees alone. (By the way, I now realise that a recent extension of these models to the case of distance-dependent networks, <http://arxiv.org/abs/1506.00348>, appeared prior to the manuscript under review here but is not cited). Now, given that the model proposed here has an extra parameter (the coupling α with the postulated metric space), and that this parameter has a special value for which no coupling is realised, it is obvious that, purely because of the presence of an extra parameter, the model can fit the data better than a model without such a parameter, but where the degrees and strengths can be equally controlled for.

We strongly disagree here. The fact that we have an extra parameter by any means implies that the model can fit the data better. Imagine that you have a model that explains some empirical observations but it fails in some others. Now you add a new mechanism that is totally opposite to the real nature of the system under study. Such model, even with more parameters, would not improve the agreement with the data. In any case, the models that you mention are not good in general at reproducing the local heterogeneity of weights, as measured by the disparity measure, or the weight distribution. This can be checked in the new set of numerical experiments that we performed in response to the second referee and that we included in the Supplementary Information in the previous resubmission.

1.A Moreover, the agreement with the empirical weight distribution is not a strong test of their model, as one would like the latter to replicate (modulo the noise) each individual weight one by one, and not the statistical distribution alone (the weight distributions of the model and the data can be identical even if no single weight is correctly replicated by the model). This leads me to the conclusion that the 6-panel figures are not conclusive about the agreement between the model and the data: simpler models without the metric hypothesis may lead to equally good results.

First notice that simpler models cannot reproduce very well the weight distribution, whereas our model is very good at this job (in fact the shape of the weight distribution is strongly dependent on the coupling, please see Supplementary Figure 2). Second, what you mention about replicating weights one by one is, in fact, related to the disparity measure, that is very well reproduced by model, as opposed to models without a metric space.

1.A 3) The only remaining real test of their hypothesis is the TIV curve. Now, I did not realise in my first reading of the manuscript that this quantity only tests the triangular inequality on the REALISED TRIANGLES in the network. It is clear that this restricts the analysis to the triples of nodes for which, at a purely topological level, the triangular inequality is already most likely to be realised. So the TIV (which is the ratio of the REALISED triangles that violate the triangular inequality to the total number of REALISED triangles) is a very weak measure of their hypothesis. I understand that the authors propose a sort of separation between the topology (which is pre-determined assuming a metric coupling) and the weights (which are established on the realised links, again assuming a metric coupling). It is however difficult to become convinced that one should not base the analysis of the violation of triangular inequality in real weighted networks to ALL the triples of nodes, including those that are not realised triangles. Clearly, if V-shaped triples of nodes are included in the analysis, the violation of triangular inequality can presumably only get much bigger, leaving us little room to believe that hidden metric spaces are indeed at play behind real weighted networks. Note that, while an analysis of V-shapes (or wedges) would be not so informative for binary networks, it would be very informative for weighted networks, given that I expect many V-shapes with two links with a strong weights (e.g. two peripheral nodes connected to the same hub) and a missing third link (between the two peripheral nodes). These weighted patterns appear to be in stark contrast with the metric hypothesis for weighted networks, and the fact that they are omitted in this analysis can be quite deceptive (again, no offence meant).

Please notice that our test of the triangle inequality is performed without the embedding of the network. Instead, we use the relation between hyperbolic distances and weights to perform it by estimating distances on the basis of observed weights. Therefore, the referee’s suggestion of using wedges is not possible because we cannot infer the distance between the two disconnected nodes.

However, we can perform a similar test to the TIV curve on wedges to check the hidden metric space hypothesis. Suppose that nodes i , j and k form a wedge in which nodes j and k are not connected. According to the hypothesis, we expect the distance x_{jk} between j and k , the disconnected pair, to be larger than the other two distances, x_{ij} and x_{ik} . Therefore, out of the three possible orderings to test the triangle inequality, the one that can be violated is $x_{ij} + x_{ik} \geq x_{jk}$. Even though we have no access to the value of x_{jk} without a value for ω_{jk} , we expect it to be larger than R because j and k are not connected and, thus,

$$x_{ij} + x_{ik} \geq x_{jk} \geq R. \quad (1)$$

Therefore, the only clear violation we can detect in wedges is

$$x_{ij} + x_{ik} < R. \quad (2)$$

In other words, assuming $x_{jk} \geq R$ for disconnected pairs, the inequality (2) implies the violation of the triangle inequality: $x_{ij} + x_{ik} < R \leq x_{jk} \Rightarrow x_{ij} + x_{ik} < x_{jk}$. However, notice that the violation of the triangle inequality does not necessarily imply inequality (2); if $x_{jk} > x_{ij} + x_{ik} \geq R$, the triangle inequality is violated but inequality (2) is not satisfied. Consequently, contrary to the referee’s intuition, if the hidden metric space hypothesis is true, not only the fraction of wedges satisfying inequality (2) should be very small for $\alpha > \alpha_{\text{real}}$, but it should also be smaller than the fraction of violations of the triangle inequality computed over topological triangles—the TIV curve.

Figure 1 shows the comparison between the TIV curve and the fraction of wedges satisfying inequality (2), as a function of α , for a synthetic network as well as for the E. Coli network. This new curve decays, in both cases, to zero at the same value of α as the TIV curve. These results confirm the conclusions presented above, yet providing further evidence of the metric nature of weights in real weighted networks.

Figure 1: Comparison of the TIV curves computed over triangles and the fraction of wedges satisfying inequality (2) computed over wedges on two networks. **Left:** Synthetic network with $\alpha = 0.5$, $\gamma = 2.5$, $\beta = 2$, $\eta = 1$, $N = 10^4$ and no noise. **Right:** E. Coli network.